# Tracking the global flows of atmospheric moisture and associated uncertainties

Obbe A. Tuinenburg[1,2,3], Arie Staal[2,3]

[1] Copernicus Institute for Sustainable Development, Utrecht University, Utrecht, 3508 TC, the Netherlands

[2] Stockholm Resilience Centre, Stockholm University, Stockholm, SE-10691, Sweden

[3] Bolin Centre for Climate Research, Stockholm, SE-10691, Sweden

*Correspondence to*: Obbe A. Tuinenburg (o.a.tuinenburg@uu.nl)

**Abstract.** Many processes in hydrology and Earth system science relate to continental moisture recycling, the contribution of terrestrial evaporation to precipitation. For example, the effects of land-cover changes on regional rainfall regimes depend on this process. To study moisture recycling, a range of moisture tracking models are in use that are forced with output from atmospheric models, but differ in various ways. They can be Eulerian (grid-based) or Lagrangian (trajectory-based), have two or three spatial dimensions, and rely on a range of other assumptions. Which model is most suitable depends on the purpose of the study, but also on the quality and resolution of the data with which it is forced. Recently, the high-resolution ERA5 reanalysis dataset has become the state-of-the-art, paving the way for a new generation of moisture tracking models. However, it is unclear how the new data can best be used to obtain accurate estimates of atmospheric moisture flows. Here we develop a set of moisture tracking models forced with ERA5 data and systematically test their performance regarding continental evaporation recycling ratio, distances of moisture flows, and 'footprints' of evaporation from seven point sources across the globe. We report simulation times to assess possible trade-offs between accuracy and speed. Three-dimensional Lagrangian models were most accurate and ran faster than Eulerian versions for tracking water from single grid cells. The rate of vertical mixing of moisture in the atmosphere was the greatest source of uncertainty in moisture tracking. We conclude that the recently improved resolution of atmospheric reanalysis data allows for more accurate moisture tracking results in a Lagrangian setting, but that considerable uncertainty regarding turbulent mixing remains. We present an efficient Lagrangian method to track atmospheric moisture flows from any location globally using ERA5 reanalysis data and make the code for this model, which we call UTrack-atmospheric-moisture, publicly available.

## 1 Introduction

Continental moisture recycling is the process whereby terrestrial evaporation re-precipitates on land, which is increasingly well understood and recognized as an important process in the Earth system. As a mechanism linking remote areas on the planet, it affects how land-cover changes influence regional precipitation (Spracklen et al., 2018), how droughts may or may not spatially propagate (Zemp et al., 2014), and whether continental interior areas receive enough precipitation for agriculture (Keys et al., 2016). With the growing interest in the topic and with increasing data availability, moisture recycling models are

being used to address a wider range of questions and on higher spatial and temporal detail. Examples include the regional hydroclimatic effects of deforestation in the Amazon (Staal et al., 2020) and the dependency of cities' water supply on upwind land areas (Keys et al., 2018).

In moisture recycling model studies, moisture is tracked through the atmosphere. This is generally done using an 'offline' model; all such models share some features, but also differ in notable ways. Universal approaches and principles among offline moisture tracking models are that they apply the atmospheric water balance (Burde and Zangvil, 2001), they run *a posteriori* using atmospheric reanalysis data or other atmospheric model output (Van der Ent et al., 2013), and at each time step, the atmospheric moisture budget is updated based on wind, evaporation, and precipitation estimates. Their output, therefore,
quantifies estimates of water transfer among any combination of locations or areas on Earth (Fig. 1).

The most notable way in which moisture tracking models differ is their representation of space. The models can be categorized into Eulerian models, which are grid-based, and Lagrangian models, which are trajectory-based. In Eulerian models, moisture flows between discrete grid cells at each time step; in Lagrangian models, individual parcels have a location with coordinates
that are updated at each time step (Fig. 2).

Besides choices regarding their grid representation (Eulerian or Lagrangian), all studies that use offline moisture tracking models make choices regarding vertical mixing of the moisture at the start of the tracking and during its path through the atmosphere, integration time step, interpolation, and resolution of the forcing dataset. In each moisture recycling study,
assumptions are chosen such that a suitable trade-off is achieved between accuracy of the representation of the downwind moisture 'footprint' of evaporation (the distribution of precipitation resulting from evaporation from a point or area), amount of data needed, and simulation time (Van der Ent et al., 2013). For example, in Eulerian models, the grid-cell size may be determined by available data, but the integration time step is not. If an explicit numerical scheme is used and the moisture flows within a single time step are much larger or smaller than the length of the grid cell, the model will give incorrect results
due to numerical inaccuracies. If the time step is chosen too large, real moisture transport may occur faster than the simulation grid and time step allow for (i.e., if the Courant number $C = \frac{v\Delta t}{\Delta x} > 1$). If the time step is taken too small, moisture transport in the model will be faster than in the forcing data. The advantage of using a Eulerian model, however, is that it is relatively fast for simulations in which moisture is released from a large fraction of the globe. The reason is that they are insensitive to an increase in scale, as all grid cells are updated with the same speed regardless of the amount of moisture present. In Lagrangian
models, a larger number of parcels released per unit of evaporation increases the computing time, making it beneficial to minimize the number of tracked parcels. However, if this number is chosen too small, the simulation is unable to capture atmospheric moisture convergence and divergence. In both Lagrangian and Eulerian models, the modeller should determine the optimal values to minimize errors.

Often, assumptions and uncertainties in moisture recycling studies are not reported. However, until now, data limitations constrained certain choices such as the minimal spatial resolution. Many recent moisture recycling studies have used ERA-Interim reanalysis data (Van der Ent et al., 2010; Van der Ent and Savenije, 2011; Tuinenburg et al., 2012; Zemp et al., 2014, 2017; Staal et al., 2018, 2020; Wang-Erlandsson et al., 2018), with a temporal resolution of six hours and a spatial resolution of 0.75° (Dee et al., 2011), as their forcing data. However, with the recent replacement of ERA-Interim by the ERA5 dataset, which has a temporal resolution of one hour and a spatial resolution of 0.25°, the trade-offs caused by the assumptions in the moisture recycling models may have shifted. The drawback of using higher-resolution data is that moisture tracking becomes more data-intensive and computing times may increase significantly. Here we assess the trade-offs and sensitivities in various atmospheric moisture recycling models forced by ERA5 reanalysis data with the aim to identify an optimal model to track the global flows of atmospheric moisture. We ask how atmospheric moisture flows can best be represented given the quality of the presently available reanalysis data. Specifically, we test the sensitivities of downwind precipitation locations to potentially important model assumptions for tracking the evaporation from seven point locations across the globe. These assumptions relate to model structure (Eulerian or Lagrangian and the number of spatial dimensions), forcing data resolution, number of tracked parcels, interpolation, and model time step. We evaluate the different model version based on a number of hydrologically relevant variables: continental evaporation recycling ratio (the percentage of evaporation that rains down over land), mean absolute latitudinal distance of the moisture transport, mean absolute longitudinal distance, and mean latitudinal and longitudinal change of the tracked moisture. We hypothesize that a Eulerian representation of the atmosphere at the resolution of ERA5 causes deviations in these variables from Lagrangian model versions. We also hypothesize that the improved resolution of vertical wind speeds allows for more accurate moisture recycling estimates, causing those estimates to deviate more with the vertical degradation of the data. Altogether, our analyses present model-dependent uncertainties in moisture recycling estimates across the globe. Based on our results we develop a moisture tracking model for ERA5 reanalysis data with optimal model assumptions and make it available on Github.

## 2 Methods

This paper tests the sensitivity in atmospheric moisture recycling to different assumptions in atmospheric moisture recycling models. In Section 2.1 we discuss the common principles of the model versions tested in this study and their differences regarding model structure and assumptions. In Section 2.2 we discuss the different simulation options that were tested.

### 2.1 Model descriptions

The offline atmospheric moisture recycling models used in this study are employed to determine the next precipitation location of evaporation that enters the atmosphere. This is done by using ERA5 atmospheric reanalysis as forcing data (Copernicus Climate Change Service), and effectively use the moisture tracking model as post-processing to this reanalysis. In general,

atmospheric moisture tracking is achieved by following moisture along its path through the atmosphere and keeping track where and how much of that moisture rains out.

The following stepwise procedure is employed to do the moisture tracking. At the starting location, a given amount of moisture enters the atmosphere through evaporation. This is the original amount of moisture to be tracked. In a Eulerian setting, this is

done on a per-grid-cell basis; in a Lagrangian setting, we track individual units that we call parcels, which is the terminology we use to describe the procedure. Once the moisture is in the atmosphere, its downwind transport is tracked using the local wind fields from the forcing data. These winds displace the parcel every time step, effectively creating a trajectory downstream from the original location. During every time step, the moisture budget over the parcel is made. Until precipitation has occurred at the location of the parcel, all the original evaporated moisture remains in the parcel. However, once there is precipitation at

the location of the parcel, a fraction of the moisture (precipitation over precipitable water of the entire atmospheric column, $\frac{P}{PW}$) that is still present in the parcel is allocated to rain out in that location. This assumes that all moisture in the atmospheric column has the same probability of raining out. Thus, the amount of original evaporation remaining decreases with downwind moisture transport. In this study, the evaporated moisture is tracked until 99% of the moisture is allocated, or the moisture has been in the atmosphere for 30 days, whichever comes first. The final step in the procedure is to determine the locations of all

the allocated moisture. The map of the allocated water represents the downwind precipitation locations of the moisture that evaporated at the starting location. We call this the downwind precipitation footprint.

Despite the commonalities between the model versions, several important assumptions are made that potentially affect the path of moisture through the atmosphere. These are discussed in the rest of this section.

**2.1.1 Eulerian and Lagrangian model versions**

Atmospheric moisture tracking models are used in either a Eulerian setting (Yoshimura et al., 2004; Dominguez et al., 2006; Van der Ent et al., 2010, 2013; Goessling and Reick, 2011; Singh et al., 2016) or a Lagrangian setting (Stohl et al., 2005; Dirmeyer and Brubaker, 2007; Tuinenburg et al., 2012). In Eulerian models, the atmosphere through which the evaporated moisture is transported is divided into grid boxes, which may be the same size as the forcing data, but may also be coarser than

the forcing data, such as in WAM-2layers (Van der Ent et al., 2014), which typically runs on 1.5° resolution. This means that moisture can only flow from one grid cell to one of its direct neighbours, which may be problematic if the time step is either too large or too small relative to the moisture flows between neighbouring cells. $C = \frac{v\Delta t}{\Delta x} > 1$

In Lagrangian models, the internal model state is not a model grid, but generally a collection of water parcels. During the

simulation, these water parcels are released and advected with the forcing wind field. The location of the parcels is not bound to the grid of the forcing data, which means that the Lagrangian model can accommodate large atmospheric moisture fluxes,

i.e. parcels can jump several grid cells of the forcing data in one model time step. The advantage of Lagrangian models is that these do not suffer from the potential numerical inaccuracies of the Eulerian models, and therefore better resemble the moisture transport in the forcing data. The simulation time of Lagrangian models scales with the number of parcels released, which is low for point releases but high if evaporation from large areas is considered.

### 2.1.2 Two-dimensional vs. three-dimensional simulations

For both the Eulerian and the Lagrangian model versions we perform simulations with two- and three-dimensional forcing data in order to test the influence of the vertical variability of atmospheric moisture flows on the moisture tracking results. When the model is forced with three-dimensional data, the horizontal (north/south, which is 'northward' in ERA5, and east/west, which is 'eastward' in ERA5) transport is driven by the wind speed at the pressure level of the parcel (Lagrangian model) or grid cell (Eulerian model). For the three-dimensional models, we distribute the released moisture over the atmospheric column according to its precipitable water content. During the simulations, there is perfect vertical mixing every 24 hours. For the Lagrangian simulations, this means that the parcel will be displaced vertically to a random altitude, weighted with the local moisture profile every 24 hours (for details, see Section 2.2.8). For the Eulerian simulations, the tracked moisture is distributed vertically proportional to the local moisture profile every 24 hours.

In case of forcing with two-dimensional data, there is no vertical variability in horizontal transport. The horizontal transport is then driven by the vertical integral of eastward and northward moisture flux (unit: $kg\ m^{-1}\ s^{-1}$) divided by the amount of moisture present in the grid cell (unit: $kg\ m^{-2}$), resulting in the average moisture flow speed (unit: $m\ s^{-1}$) for the entire atmospheric column.

### 2.1.3 Forcing data

We force the moisture tracking models with ERA5 hourly atmospheric reanalysis data on $0.25 \times 0.25°$ resolution. We use two-dimensional fields of Total Precipitation, Evaporation, Vertical integral of northward water vapour flux, Vertical integral of eastward water vapour flux and Total Column Water Vapour and three-dimensional fields of Specific Humidity, U and V components of wind speed and Vertical wind speed. For the three-dimensional fields, we use data on 25 pressure levels: every 25 hPa between 1000 hPa and 750 hPa, and every 50 hPa between 750 hPa and 50 hPa, except for the simulations with vertically degraded forcing data (see Section 2.2.5). For the Eulerian model setup, we use the same grid setup as the ERA5 forcing data, which is 0.25° spatial resolution and—for a three dimensional simulation—25 vertical layers.

## 2.2 Experimental set-up

### 2.2.1 Simulations and evaluation

We track all moisture that has evaporated from seven source locations during the first five days of July 2012. We continue to track this moisture either until it has all been allocated or the simulation reaches the end of July 2012. We perform forward tracking from point sources around the world with different climates and topographies: Chendu in China (30.75°N, 103.5°E), Central Kansas in the USA (39.0°N, 86.0°W), Manaus in Brazil (3.0°S, 60.0°W), Nagpur in India (20.0°N, 80.0°E), Nairobi in Kenya (1.25°S, 36.75°E), Stockholm in Sweden (59.5°N, 18.0°E), and Utrecht in the Netherlands (52.0°N, 5.0°E). We carry out experiments in which we evaluate the model output based on a number of criteria: visual difference in footprint, the continental recycling ratio (the percentage of evaporation that rains down over land; CRR), mean absolute latitudinal distance of the moisture transport, mean absolute longitudinal distance, and mean latitudinal and longitudinal change of the moisture. For some of the sensitivity tests, these criteria are evaluated against the simulation with the most detailed settings (most parcels, highest resolution, etc.), in which case there is a numerical true estimate. However, for some tests, there is no information to derive a true value. For these tests, the uncertainty remains higher and we derive the sensitivity of moisture recycling to the assumptions.

Despite the fact that simulation times are very much CPU-dependent, we give an estimation of the simulation time necessary (excluding the reading of the forcing data from disk). We compare the results against a baseline model that incorporates as much detail as is possible given the available data: a three-dimensional Lagrangian model with interpolated wind speeds and directions, with the high amount of released 10,000 parcels per mm of evaporation.

We show the results for Manaus in the main text and add figures for footprints of the other locations in the Supplementary Material.

Unless stated otherwise, experiments were carried out with the three-dimensional Lagrangian model version.

### 2.2.2 Number of parcels released per unit of evaporation

For the Lagrangian simulations, the trajectories of the moisture that enters the atmosphere are simulated through a number of parcels. This number should be chosen carefully. If the moisture is simulated by a small number of parcels, atmospheric moisture convergence and divergence cannot be simulated well enough. However, since the simulation times of the Lagrangian simulations scale approximately linearly with the number of parcels, simulation times may increase too much if the number of parcels chosen is too large. By default, we release 2,000 moisture parcels per mm of evaporation for all simulations. However, to test for the effect of different number of parcels, we performed simulations with 10, 50, 100, 500, 2,000, and 10,000 parcels per mm and assess their differences. Note that these numbers of parcels are on the higher side of the range of the values

typically used in moisture recycling studies (for example, Läderach and Sodemann, 2016; Sorí et al., 2017; García-Herrera et al., 2019).

### 2.2.3 Release height of moisture entering the atmosphere

We also test the differences between releasing moisture from the surface and releasing it well-mixed in the atmospheric column. Naturally, actual evaporation occurs at the surface, but moisture tracking simulations generally assume a well-mixed starting condition, which may affect the precipitation footprints of evaporation sources (Bosilovich, 2002). We test two options: either the parcels are released just above the surface, or the parcels are released at a random vertical location weighted by the local vertical humidity profile, as in (Dirmeyer and Brubaker, 2007).

### 2.2.4 Interpolation within the ERA5 space and time grid

If the internal model time step of the moisture tracking model is smaller than the one-hour temporal resolution of the ERA5 forcing data set, the forcing data need to be linearly interpolated in time. For the Lagrangian model, the same is true for the spatial grid: the parcels may be present on different locations than the grid cell centres of the forcing data. Therefore, linear interpolation on the spatial grid is done as well. As a default, all simulations in this study use linear spatial interpolation, but

this interpolation might be costly in terms of simulation time. Therefore, we test how much accuracy is lost if the simulation is run without linear interpolation and the nearest neighbour value of the forcing data is used instead.

### 2.2.5 Vertical degradation of the forcing data

The number of vertical layers can have a strong effect on the outcome (Van der Ent et al., 2013), but this has never been tested with the large amount of vertical layers (137 model levels, but output is available on 37 pressure levels, of which this study

uses 25) in ERA5. Since the size of the forcing data is substantial at full resolution (around 200 Gb for one month of forcing data), we are interested in determining the degradation of the results if we degrade the vertical resolution of the forcing data. As alternatives to non-degraded data, we consider six degradations of the forcing data, consisting of two sets of three degradations. In the first set, we use the full atmospheric column, but reduce the vertical resolution to 50 hPa ('hpa50'), 100 hPa ('hpa100') and 200 hPa ('hpa200'). In the second set, we use only forcing data between 1000 hPa and 500 hPa, since this

is where the majority of the moisture is found. The vertical resolutions of this second set are 25 hPa ('5k25'), 50 hPa ('5k50'), and 100 hPa ('5k100').

### 2.2.6 Horizontal degradation of the forcing data

Similar to the experiments with degraded vertical data, we test the effects of a degraded horizontal resolution of the forcing data. Apart from those at 0.25° resolution, we perform simulations with data on 0.5°, 1.0°, and 1.5° resolution. Instead of

interpolating the original forcing data as in the default simulations, for these simulations with horizontal degradations we average the forcing data at 0.25° to the respective degraded resolution.

### 2.2.7 Integration time step

The internal time step of the moisture tracking model can influence the simulation result. In the default simulation, it is set at 0.1 h. If it is set at a high value, the simulation time is reduced, but the Lagrangian trajectories may become unrealistic, as the forcing data are assumed to be constant during the entire time step. However, if the time step is chosen very small, simulation times might increase too much with too limited improvement in accuracy. We perform simulations with internal time steps of 0.01 h, 0.05 h, 0.1 h, 0.5 h, 1 h, 3 h and 6 h. Note that the latter two imply an effective degradation of the temporal resolution of the forcing data. For these cases we used instantaneous data on wind speed and direction.

The internal time step of the Eulerian simulations is 0.1 h, ensuring numerical stability of the simulations. In ERA5, the absolute eastward wind speed divided by the grid cell size in the east-west direction can be higher than a grid cell length per hour (for July 2012, see Fig. S1). These values are typically larger at higher latitudes than near the equator, given the smaller east-west grid cell size near the poles. Moreover, these values are larger further away from the surface, as wind speed tends to increase with altitude. The Courant numbers for vertically integrated eastward moisture transport divided by the precipitable water are generally lower than for the individual layers, but can be larger than one grid cell per hour in up to 40° of latitude away from the poles (Fig. S1). This means that if the simulation time step is too large, Courant numbers are larger than unity and moisture cannot be correctly transported on the Eulerian grid. Since decreasing the time step prohibitively increases the simulation time, Eulerian simulations were only done with a time step of 0.1 h.

### 2.1.8 Vertical displacement during transport

For the three-dimensional Lagrangian simulations, the vertical locations of the moisture parcels have to be updated during the atmospheric transport. We test several options for this vertical displacement. The first option is to use the ERA5 large-scale vertical wind speed, 'omega', for the vertical displacement. Due to all kinds of sub-grid processes, such as convection, turbulence etc., omega is almost certainly an underestimation of air mixing in the vertical direction. However, the extent to which this occurs and affects moisture recycling is unknown. Hence, apart from using omega as the input for vertical displacement, we also explore options where each parcel of moisture has a certain probability of being assigned a random new vertical position scaled by the local vertical moisture profile. This is the same procedure used to determine the initial vertical position as in Dirmeyer and Brubaker (2007). This means that moisture parcels can potentially have a quite strong vertical displacement in a short time. During every time step, there is a small probability (dt/mix-strength) of running the vertical displacement. We summarize these stochastic vertical displacement versions of the model by the mix-strength (unit: hours), or average time for one repositioning of one parcel, which is once per hour, once per six hours, once per 24 hours, and once per 120 hours. This procedure ensures that for each parcel, mixing happens on average once in the time period described by the mixing strength and that the mixing happens at random moments during the trajectory. Thus, no biases occur due to mixing at specific prescribed moments.

## 3 Results

### 3.1 Differences among Eulerian and Lagrangian models in two and three dimensions

Comparing two-dimensional and three-dimensional Eulerian and Lagrangian models, we find considerable differences in the 'footprints' of evaporation during July 2012 from our point sources across the globe. The mean difference in continental recycling ratio (CRR) with the baseline model across the source locations was 13 percentage points for the two-dimensional Eulerian model, 11 percentage points for the three-dimensional Eulerian one, 14 percentage points for the two-dimensional Lagrangian one, and zero percentage points for the three-dimensional Lagrangian one. The mean difference in absolute latitudinal transport distance with the baseline model was 4.4° for the two-dimensional Eulerian model, 4.5° for the three-dimensional Eulerian one, 2.2° for the two-dimensional Lagrangian one, and 0.0° degrees for the three-dimensional Lagrangian one. Similarly, the mean difference in absolute longitudinal transport distance was 3.2° for the two-dimensional Eulerian model, 5.3° for the three-dimensional Eulerian one, 2.2° for the two-dimensional Lagrangian one, and 0.0° degrees for the three-dimensional Lagrangian one.

Relative to the baseline model, both the two-dimensional and three-dimensional Eulerian models underestimate atmospheric transport distances in both latitudinal and longitudinal directions (only the absolute longitudinal transport in the case of Nagpur for the two-dimensional model is higher). This relatively close transport does not, however, lead to a consistent overestimation of CRR: for Nagpur, Nairobi and Utrecht the CRR is lower than in the baseline model (Table 1), which is due to geographical reasons (e.g. increased local flow from Nairobi to Lake Victoria, which is not regarded as continental in the ERA5 land mask). The simple, two-dimensional Lagrangian model tends to track moisture flows too far and therefore underestimates CRR in all simulations (Table 1). The three-dimensional Lagrangian model practically performs the same as the baseline model: all CRR estimates are equal with one percentage point accuracy. Only the absolute latitudinal distance for Utrecht and Stockholm and the absolute longitudinal distance for Stockholm were 0.1° higher than that for the baseline model (with 0.1° accuracy) (Table 1).

The case of Manaus illustrates how differences among models can cause divergent estimates for CRR (Fig. 3). The CRR varies from 38% in the two-dimensional Lagrangian model to 91% in the two-dimensional Eulerian model. The continental recycling ratios for the three-dimensional Eulerian (76%) and three-dimensional Lagrangian (68%) models are closer to that of the baseline model (68%). The high value for the two-dimensional Eulerian model coincides with its failure to simulate moisture flows across the Andes (Fig. 3A). The two-dimensional Lagrangian model simulates a relatively large flow across the Andes (Fig. 3C) followed by the three-dimensional Lagrangian model (Fig. 4D). Differences in footprints from other sources than Manaus are also substantial (Figs. S2–S7). For example, both Lagrangian models simulate a remote flow from Nairobi up to India, which is entirely absent from the simulations of the Eulerian models (Fig. S5).

Calculation times differ more among the four model versions than among the simulations in a single model version (Table 1). The simulations took (mean ± standard deviation) $2650 \pm 538$ CPU seconds with the two-dimensional Eulerian model, $20{,}470 \pm 610$ CPU seconds with the three-dimensional Eulerian model, $46 \pm 16$ CPU seconds with the two-dimensional Lagrangian model, $279 \pm 108$ CPU seconds with the three-dimensional Lagrangian model, and $1{,}384 \pm 538$ CPU seconds with the baseline model. In other words, for the point sources considered, the two-dimensional Lagrangian model is about 30 times faster than the baseline model, the three-dimensional Lagrangian model is five times faster, the two-dimensional Eulerian model is two times slower, and the three-dimensional Eulerian model is 15 times slower than the baseline model.

**3.2 Effects of number of tracked parcels**

We compared the effects of tracking different amounts of parcels in a three-dimensional Lagrangian model: 10, 50, 100, 500, 2,000 (i.e. the three-dimensional model in 3.1), and 10,000 mm⁻¹ evaporation (i.e. the baseline model). The number of parcels has a small effect on the level of detail in our case studies (although this may be different for convective events). The runs with 500 and 2,000 parcels mm⁻¹ did not result in any differences with the baseline model regarding CRR, mean absolute latitudinal distance or mean absolute longitudinal distance. The runs with 100 parcels mm⁻¹ resulted in a difference of 0.1° for both mean absolute latitudinal distance and mean absolute longitudinal distance, but no difference in CRR. The runs with 50 parcels mm⁻¹ resulted in a difference of 0.1° for both mean absolute latitudinal distance and mean absolute longitudinal distance, and a difference of one percentage point in CRR. The runs with 10 parcels mm⁻¹ resulted in a difference of 0.2° for both mean absolute latitudinal distance and mean absolute longitudinal distance, and a difference of one percentage point in CRR. It can be seen for the simulations for Manaus (Fig. 4) and the other locations (Figs. S8–S13) that the smoothness of the footprints increases with the number of tracked parcels, but the figures confirm that the patterns in the baseline model are already captured by the simulations with a relatively small amount of tracked parcels. The simulation times did differ considerably, because they scale almost linearly with the number of tracked parcels: the simulations with 2,000, 500, 100, 50, and 10 parcels mm⁻¹ were on average five times, 19 times, 83 times, 152 times, and 223 times faster than that with 10,000 parcels mm⁻¹.

**3.3 Effects of release height**

The two different ways of parcel release in the atmospheric column, moisture release at the surface and moisture release scaled with the vertical moisture profile, led to differences in evaporation footprints. Although the average difference in CRR between both model versions was zero percentage points, the model with moisture profile release produced more distant flows than that with surface release: for all locations it resulted in larger latitudinal flows, with an average of 0.2°; for all locations except Kansas it resulted in larger longitudinal flows as well, by 0.3° on average (both the mean difference and mean absolute difference).

The footprints for the simulations with moisture profile release and surface release are visually very similar (Figs. 5, S14–S19). However, the distance of moisture transport can differ substantially, as exemplified by the mean longitudinal distance of transport from Utrecht, which differed by as much as 0.8° (Fig. S19).

The average calculation time for surface release was 2% shorter than for moisture profile release.

## 3.4 Effects of interpolation

We find effects of interpolation of wind speed and direction in the three-dimensional Lagrangian model on evaporation footprints. The mean absolute difference in CRR between the interpolated and non-interpolated simulations was one percentage point. For Kansas and Nagpur, estimated CRR is lower without interpolation, but in the other cases it was higher without interpolation. Because of this lack of consistent difference, the mean CRR across locations was equal for both model versions (with an accuracy of one percentage point). The absolute latitudinal distance of moisture flows was lower without interpolation, except in the case of Kansas. Both the mean difference and mean absolute difference in latitudinal distance between the two model versions were 0.3°. The absolute longitudinal distance of moisture flows also tended to be lower without interpolation, except in the cases of Kansas and Utrecht. The mean difference in longitudinal distance was 0.1°, and the mean absolute longitudinal difference 0.3°, between the runs with the interpolated and non-interpolated data. The simulations with interpolation were on average 3% slower than those without interpolation. Visually, the differences in evaporation footprint are very small (Figs. 6, S20–S25).

## 3.5 Effects of degraded vertical atmospheric profile

The six versions of the three-dimensional Lagrangian model with a degraded vertical atmospheric profile ('hpa50', 'hpa100', 'hpa200', '5k25', '5k50', and '5k100') yielded considerable differences in evaporation footprints and their statistics: the output from 'hpa50' differed from that of the baseline model by an average of seven percentage points in CRR, by 1.7° in absolute latitudinal distance, and by 2.2° in absolute longitudinal distance. The average simulation time did not differ (with an accuracy of 1%) from that of the non-degraded model with 2,000 parcels mm$^{-1}$. The output from 'hpa100' differed from that of the baseline model by an average of ten percentage points in CRR, by 2.2° in absolute latitudinal distance, and by 3.3° in absolute longitudinal distance. The average simulation time was 1% longer than that of the non-degraded model. The output from 'hpa200' differed from that of the baseline model by an average of 17 percentage points in CRR, by 3.7° in absolute latitudinal distance, and by 4.8° in absolute longitudinal distance. The average simulation time was 5% longer than that of the non-degraded model. The output from '5k25' differed from that of the baseline model by an average of one percentage point in CRR, by 0.6° in absolute latitudinal distance, and by 0.6° in absolute longitudinal distance. The average simulation time was 4% shorter than that of the non-degraded model. The output from '5k50' differed from that of the baseline model by an average of four percentage points in CRR, by 1.0° in absolute latitudinal distance, and by 1.3° in absolute longitudinal distance. The average simulation time was 1% shorter than that of the non-degraded model. Finally, the output from '5k100' differed from

that of the baseline model by an average of nine percentage points in CRR, by 2.4° in absolute latitudinal distance, and by 3.1° in absolute longitudinal distance. The average simulation time did not differ from that of the non-degraded model.

The effects of adjustments to the vertical profile can also be seen from the evaporation footprints from Manaus, with larger flows southward and eastward (Fig. 7), and from those from the other locations (Figs. S26–S31).

### 3.6 Effects of degraded horizontal resolution

Degrading the horizontal resolution of the input data affected the results, but less than the degradation in the vertical direction did. The runs with a horizontal resolution of 0.5° differed from the baseline model by one percentage point in CRR, by 0.4° in absolute latitudinal distance, and by 0.5° in absolute longitudinal distance. The runs with a horizontal resolution of 1.0° differed from the baseline model by two percentage points in CRR, by 0.5° in absolute latitudinal distance, and by 0.6° in absolute

longitudinal distance. The runs with a horizontal resolution of 1.5° differed from the baseline model by two percentage points in CRR, by 0.7° in absolute latitudinal distance, and by 0.8° in absolute longitudinal distance. Although the patterns of the footprints are not much affected by the horizontal degradations, their level of detail is, with a progressively pixelated output as the resolution becomes coarser (Figs. 8, S32−37). The runs with horizontal degradation were on average slower than without degradation: those at 0.5° were 19% slower than those at 0.25°, those at 1.0° were 32% slower, and those at 1.5° were 76%

slower. The reason that the horizontal degradations slowed down the runs is that in this model version, the averaging of forcing data occurred during the simulations.

### 3.7 Effect of time steps

We find a low sensitivity of the footprints to reducing the time step dt. For dt = 0.01 h and dt = 0.05 h, we find no difference in CRR, absolute latitudinal distance or absolute longitudinal distance with the baseline model, which has dt = 0.1 h. For dt =

0.5 h and dt = 1 h, we find no difference in CRR, but we do find a small difference of 0.1° in mean absolute longitudinal distance and mean absolute latitudinal difference with the baseline model. For the increase in time step to dt = 3 h, we find a mean difference in CRR of one percentage point, in absolute latitudinal distance of 0.3°, and in absolute longitudinal distance of 0.4°. For dt = 6 h, we find a mean difference in CRR of two percentage points, in absolute latitudinal distance of 0.3° and in absolute longitudinal distance of 0.6°. The low sensitivity to the chosen time step is confirmed by the similarity among

footprints (Figs. 9. S38−S43).

Increasing the temporal resolution increase the running time in all cases. The runs with dt = 0.01 h were nine times slower than those with dt = 0.1 h. The ones with dt = 0.05 were two times slower, the ones with dt = 0.5 h were four times faster, and the ones with dt = 1 h, dt = 3 h, and dt = 6h were each on average seven times faster.

## 3.8 Effects of vertical mixing probabilities

Turbulence may cause considerable vertical mixing in the atmosphere, but because the rate of this mixing is unknown, there can be no baseline model to compare results against. However, we tested the sensitivity of downwind evaporation footprint to eight different rates of vertical mixing. These eight rates consist of those with and without accounting for large-scale vertical flow in the ERA5 reanalysis data (called 'omega') and of four different randomized mixing probabilities: that at which full vertical mixing takes place on average every hour, every six hours, every 24 hours, and every 120 hours.

In the simulations in which we did not account for omega, CRR decreased slightly with average mixing time (i.e. with lower mixing probability): averaged across the source locations, CRR decreases by one percentage point at each stepwise decrease in mixing probability, i.e. from hourly to six-hourly mixing, from six-hourly mixing to daily mixing, and from daily mixing to 120-hourly mixing. The absolute latitudinal transport distance increased from hourly to six-hourly mixing by an average of 0.7°. From six-hourly mixing to 24-hourly mixing this increased by another 0.2°, but from 24-hourly to 120-hourly mixing it decreased by 0.3°. The absolute longitudinal transport distance increased from hourly to six-hourly mixing by an average of 0.5°, increased from six-hourly mixing to 24-hourly mixing by 0.1°, and decreased from 24-hourly mixing to 120-hourly mixing by 0.5°. The larger spread of rainfall locations from the point sources is also clearly visible from the footprints (Figs. 10, S44–S49).

In the simulations in which we did account for omega, CRR decreased much more rapidly with increasing mixing time than in the simulations without omega: from one-hourly to six-hourly mixing, mean CRR decreased by ten percentage points, from six-hourly to daily mixing it decreased by nine percentage points, and from daily to 120-hourly mixing by four percentage points. The absolute latitudinal transport distance increased monotonically with mixing time: from hourly to six-hourly mixing by an average of 2.1°, from six-hourly to daily mixing by 0.6°, and from daily to 120-hourly mixing by 0.1°. The absolute longitudinal transport distance increased from hourly to six-hourly mixing by an average of 2.3°, increased from six-hourly mixing to 24-hourly mixing by 0.8°, and increased from 24-hourly mixing to 120-hourly mixing by 0.4°. Also here, the figures show that slower vertical mixing increases the area where rainfall depends on evaporation from the studied sources. However, with omega, the rainfall from the sources is more equally distributed within the footprints than without omega (Figs. 10, S44–S49).

CRR was higher without omega than with omega, a difference that increased with mixing time. At hourly mixing, the mean difference was four percentage points, at six-hourly mixing 13 percentage points, at daily mixing 21 percentage points, and at 120-hourly mixing 25 percentage points. Absolute latitudinal transport distance was higher with omega than without omega, and also this difference increased with mixing time: at hourly mixing, the mean difference was 1.1°, at six-hourly mixing 2.5°, at daily mixing 2.9°, and at 120-hourly mixing 3.3°. Similarly, absolute longitudinal transport distance was consistently higher

with omega than without omega: at hourly mixing, the mean difference was 1.5°, at six-hourly mixing 3.3°, at daily mixing 3.9°, and at 120-hourly mixing 4.8°.


When omega was not accounted for, the simulations ran consistently faster with larger mixing times: from hourly to six-hourly mixing by 7%, from six-hourly to daily mixing by 4%, and from daily to 120-hourly mixing by 3%. With omega included, the simulation times did not show a consistent pattern: from hourly to six-hourly mixing it decreased by 2%, but from six-hourly to daily mixing it increased by 2%, and from daily to 120-hourly mixing it increased by another 1%. Averaged across all
simulations, the calculations without omega ran 11% faster than those with omega.

## 4 Discussion

Our aim was to identify an optimal model to track the global flows of atmospheric moisture accurately and efficiently given the best available data. Therefore, we tested how different types of moisture tracking models and their assumptions, forced with the high-resolution ERA5 reanalysis data, produced different 'footprints' of evaporation from source locations. Below,
we evaluate our results and use them to propose such an optimal model, called UTrack-atmospheric-moisture, for which we publish the code.

First, we tested the performance of the two main classes of moisture tracking models, Eulerian and Lagrangian, implemented in two and three spatial dimensions. Eulerian models are grid-based, meaning that at each time step moisture is exchanged
between neighbouring grid cells. Lagrangian models track moisture parcels in continuous space, meaning that at each time step the coordinates of the parcels are updated. In both cases, the moisture budget is also updated at each time step based on the local precipitation, evaporation and precipitable water. Under the premise that tracking moisture flows becomes more accurate by processing more detailed atmospheric information, we compared our results to a three-dimensional Lagrangian model that is based on as much information as possible and tracks the very large amount of 10,000 parcels mm$^{-1}$ evaporation.
For both the Eulerian models and the two-dimensional Lagrangian model, we found large errors (>10 percentage points) in the hydrologically important variable of continental evaporation recycling ratio (CRR), which is the proportion of evaporated moisture that precipitates on land. Also, the distances of moisture transport differed by several degrees for each of these models. For many purposes, such errors are too large, while no benefit was reached regarding simulation time: both the two- and three-dimensional Eulerian models were considerably slower (58 and 73 times) than their Lagrangian equivalents. Thus, for point
sources, it can be concluded that Eulerian models, on the resolution of the ERA5 data, are not efficient compared to Lagrangian models. Although a two-dimensional Lagrangian model is fastest, the errors that result from the simplification from three to two dimensions will generally be considered too large. Therefore, we argue that an optimal moisture tracking model using ERA5 data should be a three-dimensional Lagrangian one.

We found that the accuracy of the output of the Lagrangian model was not very sensitive to the amount of parcels that were tracked for each mm of evaporation, while much simulation time can be saved by minimising that amount. At 500 parcels mm$^{-1}$, the model performance was always as good as when more parcels were tracked, so we conclude that this is a sufficiently large amount of parcels to track from point sources for one month. When the moisture is tracked from an area rather than a source, or when a longer study period is concerned, it may be justified to track even fewer parcels. This is especially the case

when one is interested in mean moisture flows rather than highly detailed spatial or temporal differences. The reason is that when a larger amount of evaporation is being considered (i.e. due to an expansion of study area or period), the total amount of parcels that is tracked can be kept equal with a corresponding reduction of amount of parcels mm$^{-1}$.

    The effect of the height in the atmospheric column at which moisture was released at the start of the simulation resulted in

small but measurable differences in the distance of the flows, where moisture release according to the vertical moisture profile yielded longer distances than moisture release at the surface. However, eventually, which model is most suitable depends on the aim of the simulation. Generally, one would want to track evaporation rather than already-present atmospheric moisture. Because evaporation occurs at the surface, and following a similar assessment by Van der Ent et al. (2013), surface release is the default in our model.


    The level of detail of the atmospheric moisture profile matters. We found that especially the degradation of this information introduced significant errors, so we advise against doing that. Similarly, our optimal model does nor reduce the horizontal resolution of the forcing data; although it affected the results to a lesser extent than reducing the vertical resolution, it affected their level of detail. In addition, interpolation of the ERA5 reanalysis data (in three spatial dimensions and the temporal

dimension) affected the footprints of evaporation, while its effect on simulation time was marginal. Therefore, we conclude that, even when all available atmospheric layers in ERA5 are used, interpolating the data is still advisable, as did previous studies using Lagrangian models (Tuinenburg et al., 2012, 2014; Van der Ent et al., 2013; Van der Ent and Tuinenburg, 2017; Tuinenburg and van der Ent, 2019).

Smaller time steps for calculation (dt) had little effect on the evaporation footprints and their statistics, but did affect the simulation times significantly. A value of dt = 0.1 h, the time step chosen in the baseline model, gave the same results as smaller values of dt, while increasing it above that did introduce some errors. Therefore, dt = 0.1 h seems optimal.

    We tested the effects of eight different assumptions on the speed of mixing in the vertical direction. In this case there is no

ideal (baseline) model to compare results with, because the real vertical mixing speed in the atmosphere is unknown, leaving only a relative comparison. Atmospheric reanalysis data do include vertical wind speeds ('omega'), but these are only grid-scale flows and do not include turbulence and convection, which have a large influence on the vertical mixing. However, while it is known that omega underestimates vertical mixing, the extent to which it does is not. Therefore, moisture tracking models

may complement the omega-based vertical displacement by using a mixing scheme based on turbulent mixing (Stohl et al., 2005), or disregard omega and either choose another method to account for vertical mixing, such as based on transporting parcels on isentropic levels (Dirmeyer and Brubaker, 2007), or distribute the moisture budget errors over the vertical layers (WAM-2layers; Van der Ent et al., 2014). In our case, at each time step each moisture parcel had a certain probability of being relocated within the atmospheric column (randomly, but scaled with the humidity content of the atmosphere). These probabilities were chosen such that on average every hour, six hours, 24 hours, or 120 hours full mixing will have occurred. All four options were assessed with and without additionally accounting for large-scale vertical wind speed. We found that whether or not omega was accounted for had a large effect on the results. This is to be expected especially when many vertical layers are used between which flows take place. The effect of omega declines with more rapid randomized mixing. Because of the uncertainties related to vertical mixing, we leave the mixing time and inclusion of omega as an option in the model. As default we take a mixing time of 24 h without omega. The rationale for choosing 24 h is twofold: first, it is within the range (6−24 h) where the results are relatively robust to the choice of mixing time; second, atmospheric mixing follows a diurnal cycle (Tuinenburg and van der Ent, 2019), which is averaged out by mixing continuously with a full mixing every 24 hours. While we assume that this speed of mixing is rapid enough to supersede larger-scale vertical flows so as to simplify the model and exclude omega, the moisture tracking results can be very sensitive to vertical mixing. This study does not use a true reference to compare the vertical mixing assumptions against. This means that it is hard to justify a definitive choice. Experiments using tracers could be devised to constrain the uncertainty regarding the vertical transport further and may lead to inclusion of omega or a more physically based mixing.

The uncertainty regarding the process of vertical mixing combined with the sensitivity of moisture recycling to the vertical mixing introduces a corresponding uncertainty in any moisture recycling result. Based on our sensitivity analysis, it can be expected that in this respect alone, the uncertainty in transport distance is limited to several tenths of degrees in both latitudinal and longitudinal directions. Continental recycling ratios differed only by a few percentage points depending on mixing assumptions. In case randomized mixing, representing turbulence, would need to be complemented by omega, the uncertainty becomes very sensitive to the level of turbulence. We found that continental recycling ratios could differ in the range of ten percentage points and transport distances by several degrees. These are large uncertainties, so we argue that a better constraint of vertical mixing could greatly improve the accuracy of moisture tracking models in the future.

We used a selection of point sources and tracked the destinations of their evaporation for a single arbitrary month (July 2012). Although moisture recycling can strongly vary both spatially and temporally (Brubaker et al., 1993; Gimeno et al., 2012), we chose to focus on the spatial variation to cover a range of terrains, latitudes, and wind patterns, as the effects of different model assumptions may become apparent especially by studying different regions (Goessling and Reick, 2013). Here the purpose was to compare the performance of different models and their assumptions for a range of hydrological statistics as well as

simulation time. Thus, the footprints may differ in their representativeness for the respective locations, but they are helpful in visualising the consequences of different assumptions in atmospheric moisture tracking.

Although we do a number of sensitivity analyses regarding important assumptions related to atmospheric moisture tracking, we used only one moisture tracking model. It should be noted that more atmospheric tracer models such as HYSPLIT (Stein et al., 2015) and moisture recycling models (Stohl et al., 2005; Van der Ent et al., 2014; Döös et al., 2017; Keune and Miralles, 2019) exist that could be used with ERA5 forcing. Comparison with the tracer models would require keeping track of the atmospheric moisture balance during tracking. Here, we focus on the process uncertainties in combination with the use of

ERA5 reanalysis data. These other models currently use different forcing datasets. Therefore, it would be difficult to draw conclusions at the process level when results are compared, because it is hard to correct for the different forcing datasets. We would welcome a moisture tracking model intercomparison in which forcing dataset and case study are prescribed.

Many moisture recycling studies have used low-spatial-resolution Eulerian models (typically 1.5° with one or two vertical

layers) forced with ERA-Interim reanalysis data (Dee et al., 2011). When Eulerian models are run on coarse spatial scales, the risk of numerical instability due to large Courant numbers is contained. However, we recommend caution when developing Eulerian models based on the resolution of ERA5 data, because of the large moisture fluxes compared to the grid cell size and the numerical dispersion when this is mitigated by using smaller time steps. Downgrading the data to coarser resolutions such as 1.5° would circumvent the problem, but comes at the cost of loss of information. The Lagrangian model we present here

does not suffer from numerical issues caused by a grid-based approach, and can be expected to become more accurate with the availability of more accurate atmospheric data.

In this study, we track the moisture for up to 30 days. It should be noted that the accuracy of moisture tracking results decreases with tracking time (Sodemann and Stohl, 2009). This is due to the fact that moisture convergence and divergence patterns in

the forcing dataset are well represented if there are many parcels present around that location. However, many days after the release of the parcel, it will be transported far away from the release location. Therefore, the density of the parcels released from a specific location will decrease with time and also decrease the moisture tracking accuracy. This is the reason many moisture tracking models stop their simulations after ten days. We chose not to do this, because although in many cases almost all the moisture has been allocated within ten days (Sodemann, 2020), unfortunately, in other cases only a small fraction of

moisture has been allocated after ten days. Nevertheless, we recognize that the tracking accuracy decreases with simulation time.

The facts that the errors in moisture recycling estimates depend on the study area and that we tracked only moisture for one month that evaporated during five days mean that one should be cautious with generalizing the implications of these outcomes.

An example of moisture flow above complex terrain is that from Manaus in the Amazon, where westward moisture flows are

blocked by the Andes and partially diverted southward, but also partially pass over the mountain range and precipitate over the Pacific Ocean. We argue that for areas with relatively complex terrain in particular, three-dimensional Lagrangian models are most suitable because these describe atmospheric moisture transport better under situations with strong vertical variability in horizontal moisture transport.

**5 Conclusions**

Moisture recycling science has a long history, with gradually improved models depending on the state-of-the-art data, from early one-dimensional work to explicit moisture tracking in two-dimensional and three-dimensional Eulerian and Lagrangian models. This model development has gone hand-in-hand with improved data development. With the development of the new ERA5 reanalysis data, data limitations have decreased considerably. We evaluated the performance of different model types 550 given these new data. Our comprehensive sensitivity analysis led us to propose an optimal three-dimensional Lagrangian moisture tracking model – one that is able to produce highly detailed 'footprints' of evaporation and that is devoid of unnecessary complexity. It therefore runs relatively fast with negligible loss of information. Furthermore, we conclude that in our and any other moisture tracking method, the vertical mixing assumptions are the most significant. Therefore, we recommend to focus further research on this vertical moisture transport. Finally, we make the code for our model freely 555 available.

**Code and data availability**

The model code is available at: https://github.com/ObbeTuinenburg/UTrack-atmospheric-moisture. The ERA5 data are available at: https://cds.climate.copernicus.eu/cdsapp#!/home.

**Author contributions**

OAT conceived and designed the study. Both authors carried out the study, interpreted the results and wrote the paper.

**Acknowledgments**

We thank Ruud van der Ent, Ingo Fetzer, Line Gordon and Lan Wang-Erlandsson for useful discussions. We thank both reviewers and Jolanda Theeuwen for helpful comments on the manuscript. We thank the Bolin Centre for Climate Research for funding the stay of OAT at the Stockholm Resilience Centre. OAT acknowledges support from the research program 565 Innovational Research Incentives Scheme Veni (016.veni.171.019), funded by the Netherlands Organisation for Scientific Research (NWO). AS acknowledges support from the European Research Council project Earth Resilience in the Anthropocene (743080 ERA).

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

**Tables and Figures**

Table 1: Continental recycling ratio (%), absolute latitudinal distance (°), absolute longitudinal distance (°), mean latitudinal change (°), mean longitudinal change (°), and calculation time (CPU seconds) for the baseline, two-dimensional Eulerian, three-dimensional Eulerian, two-dimensional Lagrangian, and three-dimensional Lagrangian model versions for each of the seven point sources in July 2012. For continental recycling ratio, absolute latitudinal distance, absolute longitudinal distance, mean latitudinal change, and mean longitudinal change, the absolute differences with the baseline model are also given.

| | Baseline | 2D Eulerian | 3D Eulerian | 2D Lagrangian | 3D Lagrangian |
|---|---|---|---|---|---|
| Continental recycling ratio (%) | | | | | |
| Chendu | 94 | 99 | 96 | 72 | 94 |
| Kansas | 44 | 51 | 47 | 38 | 44 |
| Manaus | 68 | 91 | 76 | 38 | 68 |
| Nagpur | 94 | 78 | 80 | 77 | 94 |
| Nairobi | 87 | 73 | 62 | 75 | 87 |
| Stockholm | 63 | 83 | 80 | 51 | 63 |
| Utrecht | 45 | 35 | 34 | 42 | 44 |
| Mean diff. | − | 13 | 11 | 14 | 0 |
| Absolute latitudinal distance (°) | | | | | |
| Chendu | 2.6 | 1.1 | 1.0 | 5.8 | 2.6 |
| Kansas | 7.2 | 1.7 | 2.8 | 7.6 | 7.2 |
| Manaus | 4.8 | 0.8 | 1.4 | 6.6 | 4.8 |
| Nagpur | 5.1 | 1.7 | 1.2 | 5.9 | 5.1 |
| Nairobi | 7.3 | 4.0 | 2.3 | 14.8 | 7.3 |
| Stockholm | 7.4 | 2.6 | 2.5 | 8.1 | 7.5 |
| Utrecht | 11.4 | 2.9 | 3.1 | 12.4 | 11.5 |
| Mean diff. | − | 4.4 | 4.5 | 2.2 | 0.0 |
| Absolute longitudinal distance (°) | | | | | |
| Chendu | 3.4 | 0.8 | 1.7 | 6.2 | 3.4 |
| Kansas | 19.6 | 9.2 | 9.3 | 20.6 | 19.6 |
| Manaus | 15.0 | 8.6 | 7.5 | 14.3 | 15.0 |
| Nagpur | 5.1 | 5.7 | 4.3 | 6.6 | 5.1 |
| Nairobi | 6.1 | 1.9 | 2.4 | 11.0 | 6.1 |
| Stockholm | 15.0 | 7.0 | 6.0 | 16.8 | 15.1 |

| | | | | | |
|---|---|---|---|---|---|
| Utrecht | 16.0 | 13.7 | 11.7 | 18.2 | 16.0 |
| Mean diff. | − | 3.2 | 5.3 | 2.2 | 0.0 |

| Mean latitudinal change (°) | | | | | |
|---|---|---|---|---|---|
| Chendu | 2.3 | 0.6 | 0.1 | 5.6 | 2.3 |
| Kansas | 4.0 | 1.3 | 1.8 | 4.5 | 4.0 |
| Manaus | 1.5 | 0.4 | 0.0 | 4.3 | 1.5 |
| Nagpur | 4.9 | 1.3 | 1.3 | 5.8 | 4.9 |
| Nairobi | 7.2 | 4.0 | 2.2 | 14.8 | 7.2 |
| Stockholm | 1.3 | 2.6 | 2.3 | 0.0 | 1.3 |
| Utrecht | 6.1 | 2.8 | 3.1 | 8.5 | 6.0 |
| Mean diff. | − | | | | |

| Mean longitudinal change (°) | | | | | |
|---|---|---|---|---|---|
| Chendu | 2.4 | 0.6 | 0.9 | 6.0 | 2.4 |
| Kansas | 15.2 | 9.2 | 8.5 | 14.6 | 15.2 |
| Manaus | −14.3 | −8.6 | −7.4 | −14.3 | −14.4 |
| Nagpur | 3.6 | 5.7 | 3.8 | 5.2 | 3.6 |
| Nairobi | 3.0 | −1.6 | 0.8 | 6.2 | 3.0 |
| Stockholm | 11.1 | −0.6 | −1.4 | 13.3 | 11.1 |
| Utrecht | 15.3 | 11.3 | 9.8 | 17.9 | 15.4 |
| Mean diff. | − | 5.1 | 4.8 | 2.0 | 0.0 |

| Calculation time (CPU seconds) | | | | | |
|---|---|---|---|---|---|
| Chendu | 821 | 2612 | 20366 | 35 | 166 |
| Kansas | 2182 | 2601 | 20229 | 69 | 439 |
| Manaus | 1553 | 2670 | 21713 | 53 | 313 |
| Nagpur | 808 | 2655 | 20275 | 25 | 163 |
| Nairobi | 924 | 2663 | 20733 | 31 | 186 |
| Stockholm | 1623 | 2611 | 20140 | 52 | 327 |
| Utrecht | 1774 | 2741 | 19836 | 57 | 357 |


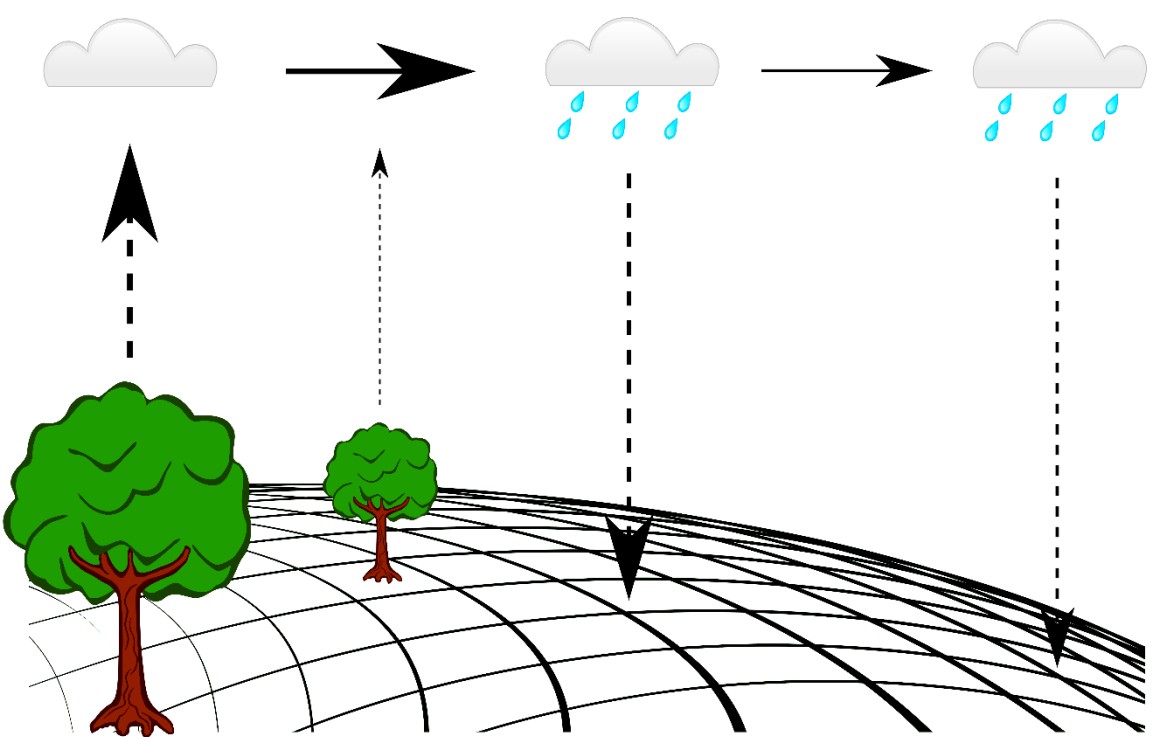

**Figure 1: Moisture tracking from source to sink. All moisture tracking models use atmospheric reanalysis data to simulate the locations of moisture. At each time step, moisture budgets are updated based on wind speed and directions (horizontal arrows), evaporation (dashed arrows up) and precipitation (dashed arrows down). This leads to source-to-sink estimates of atmospheric moisture flows, such as the evaporation footprints and basin recycling ratios in this study.**

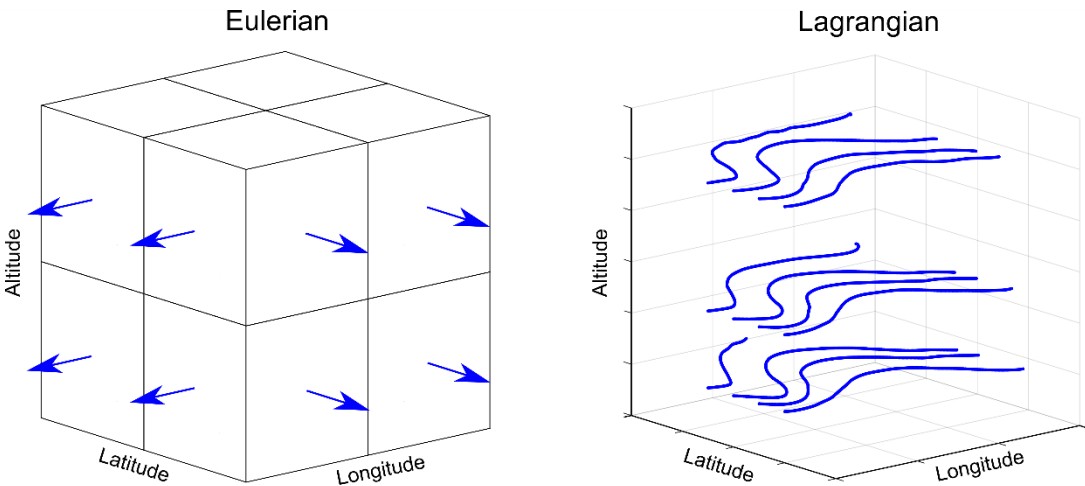

Figure 2: The difference between Eulerian and Lagrangian moisture tracking models. A) Eulerian models are grid-based, meaning that the study area is divided into a two- or three-dimensional grid of cells. At each time step, the tracked moisture content of each grid cell is updated based on estimated cell-to-cell winds, precipitation, and evaporation. B) Lagrangian models are trajectory-based, meaning that a number of moisture parcels have coordinates. At each time step, the coordinates and the tracked moisture content of the parcels are updated based on point-based wind flows, precipitation, and evaporation.

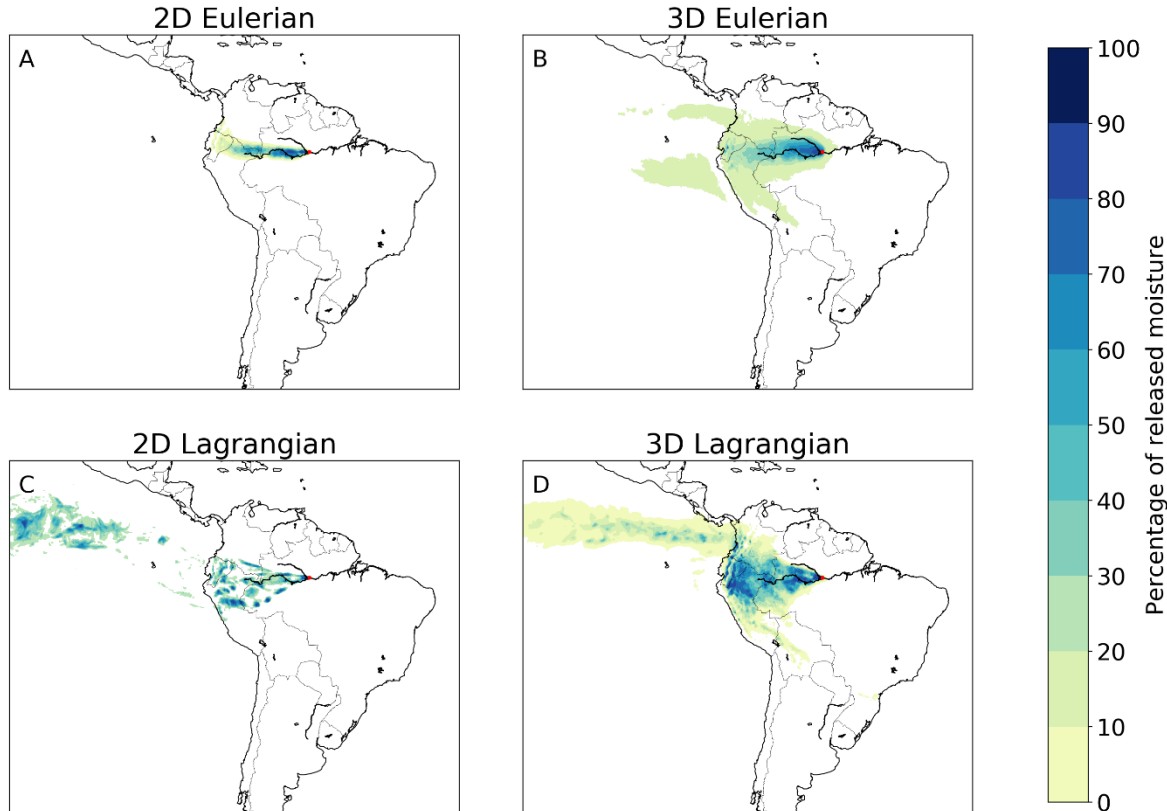

**Figure 3: Different footprints of moisture releases from Manaus in July 2012 in two-dimensional and three-dimensional Eulerian and Lagrangian models. A) Two-dimensional Eulerian, with a mean latitudinal moisture flow of 0.4° in northerly direction and mean longitudinal flow of 8.6° in westerly direction; B) Three-dimensional Eulerian, with a mean latitudinal moisture flow of 0.0° in northerly/southerly direction and mean longitudinal flow of 7.4° in westerly direction; C) Two-dimensional Lagrangian, with a mean latitudinal moisture flow of 4.3° in northerly direction and mean longitudinal flow of 14.3° in westerly direction; D) Three-dimensional Lagrangian, with a mean latitudinal moisture flow of 1.5° in northerly direction and mean longitudinal flow of 14.4° in westerly direction.**


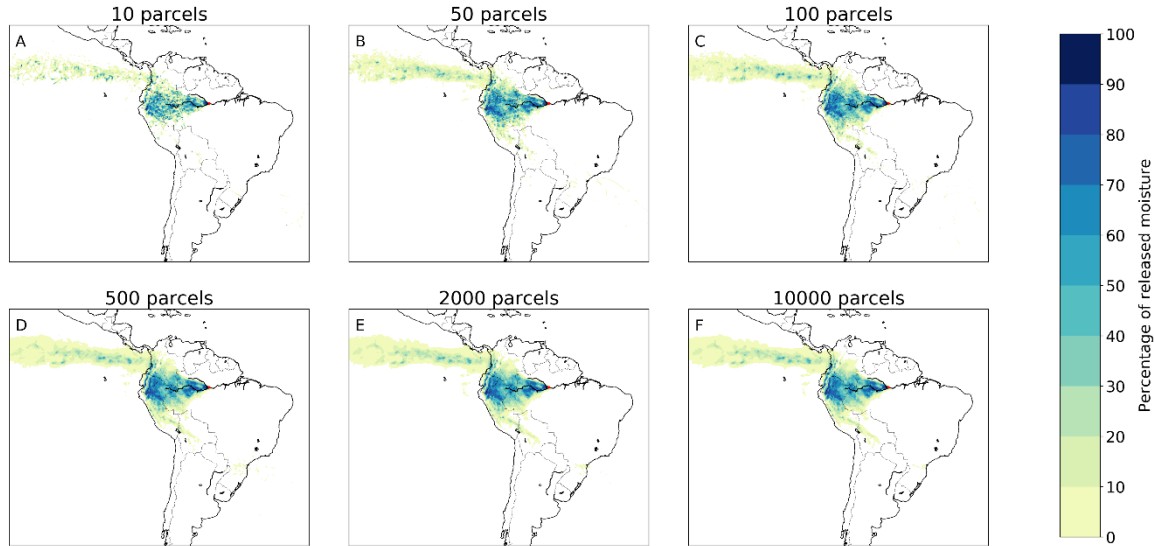


**Figure 4: Different footprints of moisture releases from Manaus in July 2012 in a three-dimensional Lagrangian model with 10, 50, 100, 500, 2,000, and 10,000 tracked parcels mm[-1]). A) 10 parcels, with a mean latitudinal moisture flow of 1.3° in northerly direction and mean longitudinal flow of 14.2° in westerly direction; B) 50 parcels, with a mean latitudinal moisture flow of 1.6° in northerly direction and mean longitudinal flow of 14.3° in westerly direction; C)**

**100 parcels, with a mean latitudinal moisture flow of 1.4° in northerly direction and mean longitudinal flow of 14.2° in westerly direction; D) 500 parcels, with a mean latitudinal moisture flow of 1.4° in northerly direction and mean longitudinal flow of 14.3° in westerly direction; E) 2,000 parcels, with a mean latitudinal moisture flow of 1.5° in northerly direction and mean longitudinal flow of 14.4° in westerly direction; F) 10,000 parcels, with a mean latitudinal moisture flow of 1.5° in northerly direction and mean longitudinal flow of 14.3° in westerly direction.**


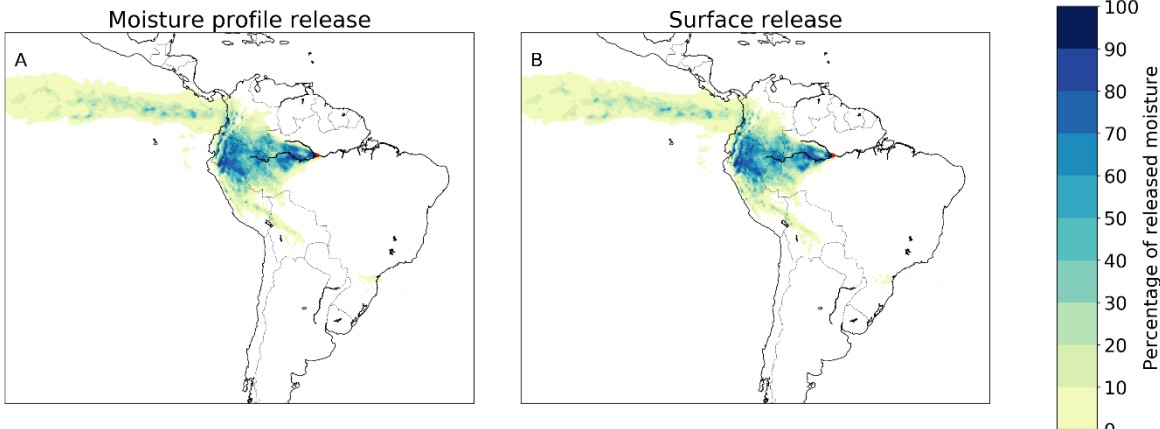

**Figure 5: Different footprints of moisture releases from Manaus in July 2012 in a three-dimensional Lagrangian model**
**with moisture released according to the vertical moisture profile of the atmosphere and moisture released at the surface.**
**A) Release according to the moisture profile, with a mean latitudinal moisture flow of 1.5° in northerly direction and**
**mean longitudinal flow of 14.4° in westerly direction; B) Release at the surface, with a mean latitudinal moisture flow**
**of 1.3° in northerly direction and mean longitudinal flow of 14.2° in westerly direction.**


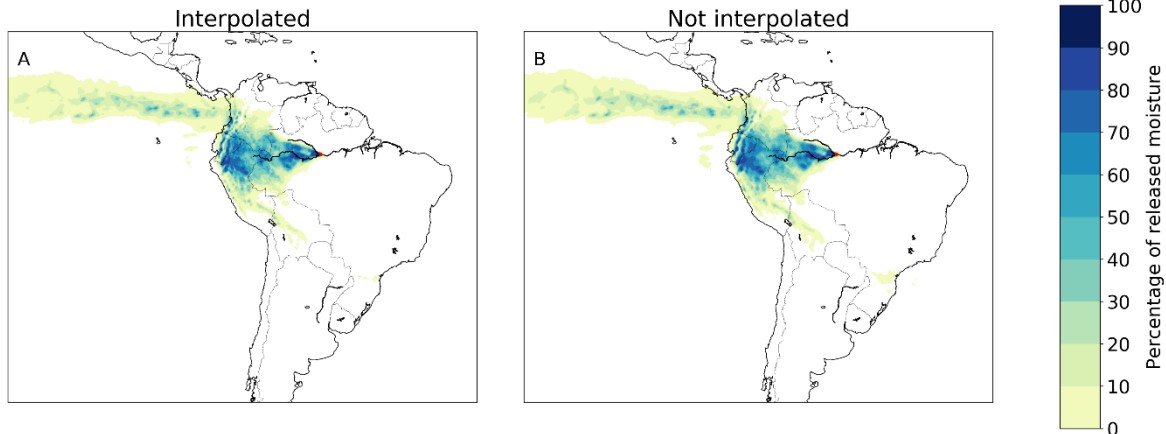


**Figure 6: Different footprints of moisture releases from Manaus in July 2012 in a three-dimensional Lagrangian model with and without interpolation of wind speed and directions. A) Interpolated, with a mean latitudinal moisture flow of 1.5° in northerly direction and mean longitudinal flow of 14.4° in westerly direction; B) Not interpolated, with a mean latitudinal moisture flow of 1.4° in northerly direction and mean longitudinal flow of 14.3° in westerly direction.**


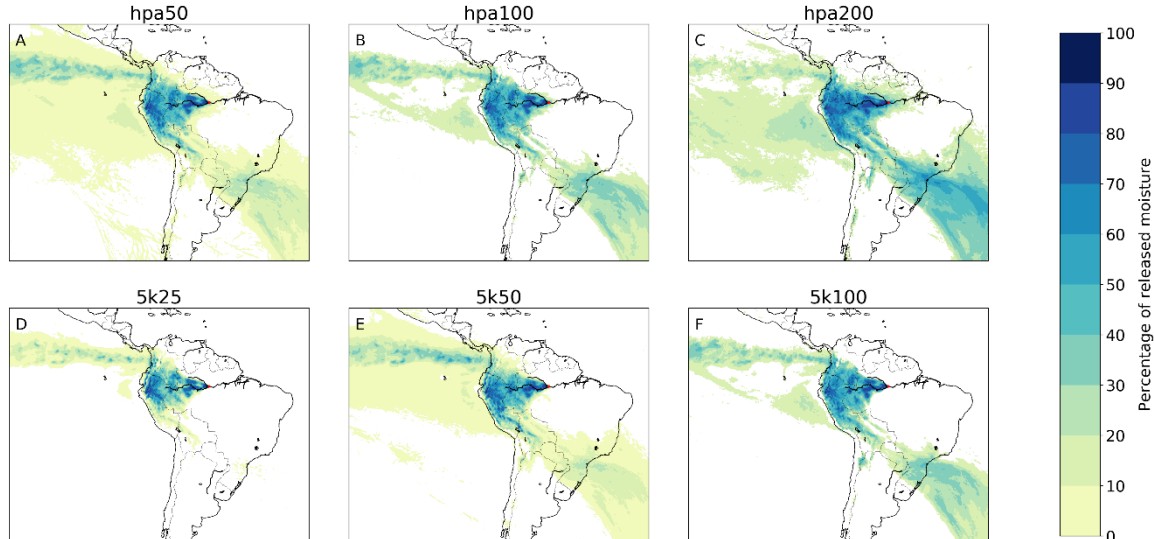

**Figure 7: Different footprints of moisture releases from Manaus in July 2012 in a three-dimensional Lagrangian model with different degradations of the vertical moisture profile. A) hpa50, with a mean latitudinal moisture flow of 2.8° in southerly direction and mean longitudinal flow of 11.0° in westerly direction; B) hpa100, with a mean latitudinal moisture flow of 5.3° in southerly direction and mean longitudinal flow of 9.2° in westerly direction; C) hpa200, with a mean latitudinal moisture flow of 17.6° in southerly direction and mean longitudinal flow of 1.8° in westerly direction; D) 5k25, with a mean latitudinal moisture flow of 1.7° in northerly direction and mean longitudinal flow of 14.4° in westerly direction; E) 5k50, with a mean latitudinal moisture flow of 2.8° in southerly direction and mean longitudinal flow of 11.6° in westerly direction; F) 5k100, with a mean latitudinal moisture flow of 5.1° in southerly direction and mean longitudinal flow of 9.6° in westerly direction.**

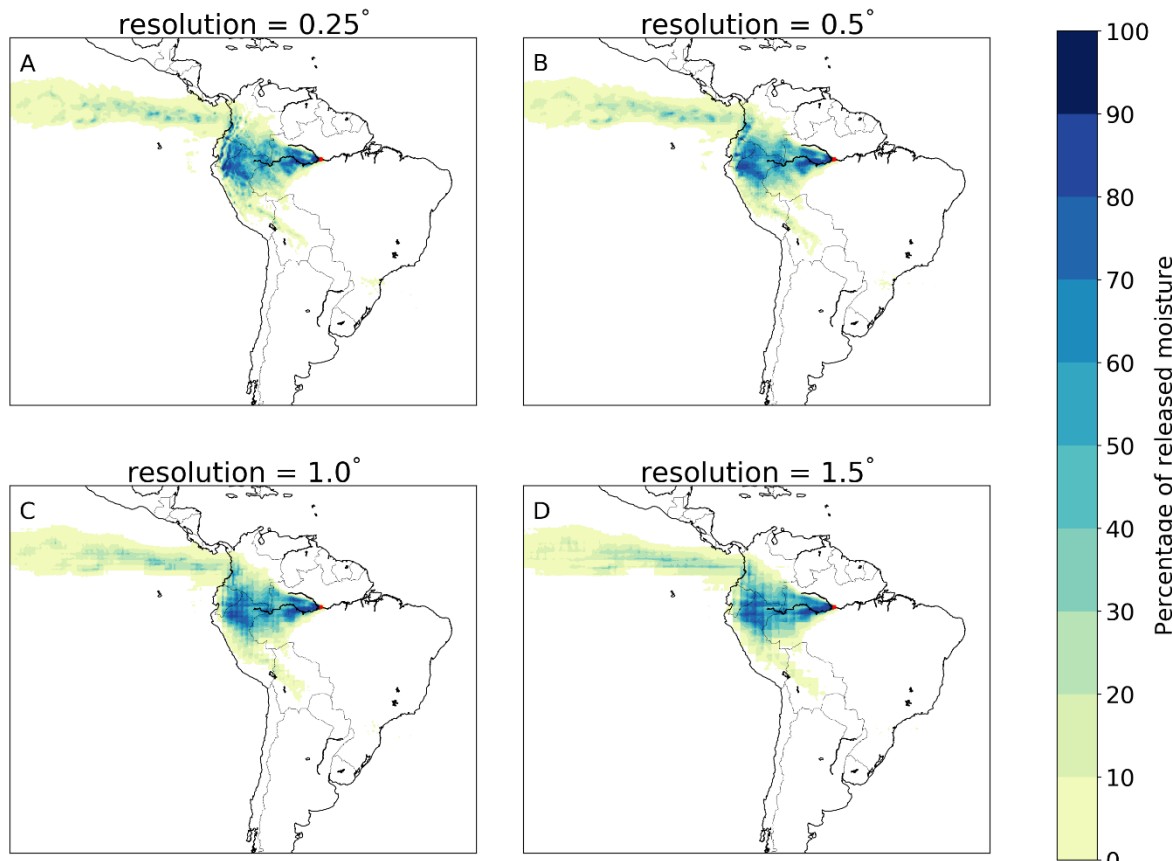

**Figure 8: Different footprints of moisture releases from Manaus in July 2012 in a three-dimensional Lagrangian model**
**with different horizontal resolutions. A) 0.25°, with a mean latitudinal moisture flow of 1.5° in northerly direction and mean longitudinal flow of 14.4° in westerly direction; B) 0.5°, with a mean latitudinal moisture flow of 1.6° in northerly direction and mean longitudinal flow of 14.3° in westerly direction; C) 1.0°, with a mean latitudinal moisture flow of 1.7° in northerly direction and mean longitudinal flow of 14.2° in westerly direction; D) 1.5°, with a mean latitudinal moisture flow of 1.7° in northerly direction and mean longitudinal flow of 14.2° in westerly direction.**

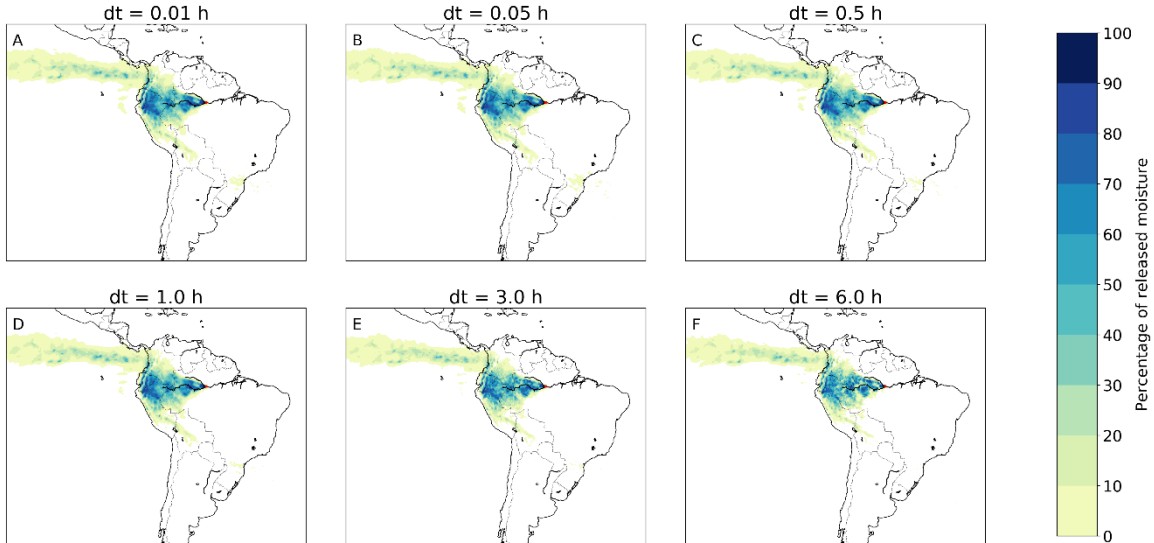


**Figure 9: Different footprints of moisture releases from Manaus in July 2012 in a three-dimensional Lagrangian model with different time steps (dt): 0.01 hours, 0.05 hours, 0.5 hours, 1.0 hours, 3.0 hours, and 6.0 hours. A) 0.01 h, with a mean latitudinal moisture flow of 1.4° in northerly direction and mean longitudinal flow of 14.3° in westerly direction; B) 0.05 h, with a mean latitudinal moisture flow of 1.3° in northerly direction and mean longitudinal flow of 14.2° in**

**westerly direction; C) 0.5 h, with a mean latitudinal moisture flow of 1.5° in northerly direction and mean longitudinal flow of 14.4° in westerly direction; D) 1.0 h, with a mean latitudinal moisture flow of 1.5° in northerly direction and mean longitudinal flow of 14.5° in westerly direction; E) 3.0 h, with a mean latitudinal moisture flow of 1.6° in northerly direction and mean longitudinal flow of 14.3° in westerly direction; F) 6.0 h, with a mean latitudinal moisture flow of 1.7° in northerly direction and mean longitudinal flow of 14.2° in westerly direction.**

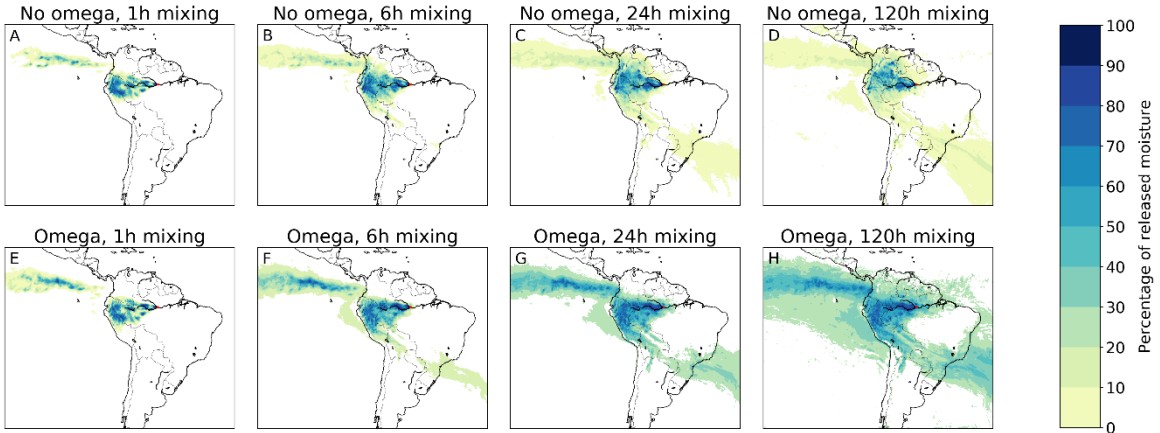


**Figure 10: Different footprints of moisture releases from Manaus in July 2012 in a three-dimensional Lagrangian model with different mixing assumptions: without and with accounting for the three-dimensional moisture flows in the ERA5 data (termed omega), and with different assumptions of additional vertical mixing speed (full mixing every 1 h, every 6 h, every 24 h, and every 120 h). A) Without omega, every 1 h mixing, with a mean latitudinal moisture flow of 2.4° in**
**northerly direction and mean longitudinal flow of 16.4° in westerly direction; B) Without omega, every 6 h mixing, with a mean latitudinal moisture flow of 1.4° in northerly direction and mean longitudinal flow of 14.3° in westerly direction; C) Without omega, every 24 h mixing, with a mean latitudinal moisture flow of 0.1° in northerly direction and mean longitudinal flow of 11.6° in westerly direction; D) Without omega, every 120 h mixing, with a mean latitudinal moisture flow of 1.2° in northerly direction and mean longitudinal flow of 9.0° in westerly direction; E) With**
**omega, every 1 h mixing, with a mean latitudinal moisture flow of 3.0° in northerly direction and mean longitudinal flow of 16.7° in westerly direction; F) With omega, every 6 h mixing, with a mean latitudinal moisture flow of 0.3° in northerly direction and mean longitudinal flow of 14.5° in westerly direction; G) With omega, every 24 h mixing, with a mean latitudinal moisture flow of 5.0° in northerly direction and mean longitudinal flow of 12.3° in westerly direction; H) With omega, every 120 h mixing, with a mean latitudinal moisture flow of 6.9° in northerly direction and mean**
**longitudinal flow of 11.5° in westerly direction.**