# Peer review of "Tracking the global flows of atmospheric moisture and associated uncertainties"

_Hydrology and Earth System Sciences, 2019_

## Referee Comment (RC1) · Munir Nayak (Referee) · 17 Dec 2019

Review of paper titled "Tracking the global flows of atmospheric moisture" submitted to Hydrology and Earth System Sciences,

Manuscript Number: hess-2019-597

Dear Dr. Louise

Thank you for sending the manuscript to me for a review. Below you will see a short summary of the manuscript, followed by my specific (major) comments, and then technical corrections (minor comments) at the end.

**Summary**

Tracking moisture in the atmosphere over time has many applications in the fields of hydrology and meteorology, such as finding the major moisture sources of particular extreme precipitation event at a given location. Moisture tracking models can be represented with a variety of schemes, which include Eulerian and Lagrangian (in two and three dimensions) frameworks, different integration time steps, different sets of vertical forcings, representations of vertical wind velocities, locations of moisture releases to the atmosphere, etc. The results of moisture tracking, for example evaporation recycling rate, the distance travelled by moisture, etc., will depend on the scheme chosen. The authors experiment a set of evaporation tracking schemes to asses the sensitivity of tracking results to different schemes. In summary, the steps used in the manuscript can be written as: a.) Select seven point-sources of evaporation across the globe, b.) track the evaporation (during first five days of July 2012) from these locations using a tracking scheme, c.) keep track of precipitation locations, i. e., latitude and longitude points where precipitation happened, d.) repeat the above three steps with different model settings, and e) compare the tracking results. Based on the comparisons among different schemes, the authors propose an "optimal" tracking scheme for general hydrological applications: 3D-

Munir Ahmad Nayak

Lagrangian, 500 particles released per mm evaporation, moisture releases at surface, linear interpolation in time and space, adding as many vertical forcings as possible, etc.

The manuscript is very clearly written, and overall, I think it can be of interest to many readers of HESS and other similar journals. There, however, are some places in the manuscript where authors should provide more justifications and clarifications; I added these in the "specific comments" section below. I hope the authors can address these comments, after which the manuscript may be suitable for publication in HESS. I will be looking forward to reading a revised draft.

**Specific comments**

1. When an air parcel moves, it gains and losses moisture along its track, the gain and loss can be attributed directly to the location where the change happens if the parcel is within the boundary layer. When the parcel is out of the boundary layer, the change locations are not clearly evident, since they can come from remote sources (Sodemann et al., 2008), which are difficult to evaluate. It is not clear how the "original" evaporation (Lines 95-96) is maintained throughout the parcel's track. More clarity on this will help in interpreting many results presented in section 3, for example evaporation footprint.

2. Section 2.1.4: During convective up- and down drafts, horizontal winds also show significant changes in magnitude and direction; the particles can then be displaced vertical depending on the changes in the vertical winds, instead of assigning random vertical displacements to them, which seems arbitrary. If feasible, another scheme based on this large horizontal wind gradients may be added in the present framework.

3. The basic structure of the model is not presented anywhere. I suggest adding a stepwise procedure on how the tracking is performed. Actually, response to this might answer my first comment 1 also.

Munir Ahmad Nayak

4. This baseline model is 3D Lagrangian with 10,000 parcels released per mm; the 3D model in L243, table 1, and other results almost identical to the baseline model. This does not seem a reasonable way to compare models and present results, since baseline itself is not "True Tracking" and cannot be a perfect reference. It might be a good idea to use other models' output as reference, such as HYSPLIT (Draxler and Hess, 1998), LAGRANTO (Wernli, H., and H. C. Davies, 1997).

5. In Section 3.2, it is argued that number of parcels released does not affect tracking results greatly. We should note, however, that number of parcels may matter to capture convective/converging and diverging events, as stated by the authors in section 2.2.4. Here, the simulations are run only for one case (July 2012), which may not have large convergence or divergence at any time. We should be careful in generalizing these results to all events, unless simulations results of some specific convective events show similar results.

**Technical corrections**

Define "footprints" at the beginning, somewhere in the introduction.

One of the aims of the manuscript was to evaluate model structure; however, it is not clear where model structures have changed. Perhaps, Eulerian and Lagranian can be taken as different model structures, but this needs to be written explicitly.

L25-26: Fig. 1 does not specifically show moisture recycling as indicated here.

L46-48: Rather than "assumptions", I feel they are more like user "choices".

L44: Here, I suggest writing "parcels" instead of "particles".

L51-54: It is not clear how the results will be incorrect; also, clearly explain why the Eulerian model simulations will not be as fast as Lagrangian when moisture is released from small areas.

L60: Do you mean "which resulted in Courant numbers exceeding one …"?

Munir Ahmad Nayak

L125: I am not sure if I understand why vertical mixing is to be carried out every time interval and how is it performed; more details on this can help readers.

L130-L133: Rephrase for more clarity.

L150: Do you mean "particle" instead of "parcel"? Try to be consistent.

L153: No, this does not seem realistic; you might not be able to capture convergence or divergence with this scheme, just because it is random.

L178: Here, 10,000 particles are released per mm of evaporation over first 5 days of July 2012? Evaporation from a point source at any instant will be transported during each time step; are we releasing parcels at one instant, say t=0, or over multiple time steps (t =0, t=1, and so on.). Add a few lines to clearly explain how parcels are released, and how evaporation over 5 days will be captured by parcels released.

L189: In table 1: I would also add a simple metric "mean distance travelled".

L225-234: The entire section can be as a separate row in Table 1.

L250: Low value of CRR is observed in 2D Lagrangian case, not Eulerian; see Figure 3c also

L254: Fig. S5, not S4.

L259-262: Give clear reasons of so much computational time difference between 3D Lagrangian and 3D Eulerian schemes. In other words, why do we think 3-D Eulerian takes so much time?

Figures 6, 7 and other similar figures: Since these figures do not shown clear differences, perhaps it is better to show differences directly, i.e., map of baseline footprint minus footprint from the given set up.

Figure S25B: There is an unusual straight color line in this panel; can that be removed?

Section 3.5: Were the models run with interpolation or with nearest neighbor method. Also, the results here can be concisely presented in a tabular form, rather than text.

Section 3.6: Will it be feasible to test sensitivity to timestep dt = 3 hours?

Munir Ahmad Nayak

Section 3.7: I am not sure if I clearly understand the purpose of mixing and its usefulness in practice. Perhaps provide more details.

**References**

1. Sodemann, H., C. Schwierz, and H. Wernli, 2008: Interannual variability of Greenland winter precipitation sources: Lagrangian moisture diagnostic and North Atlantic Oscillation influence. *J. Geophys. Res.*, **113**, D03107.

2. R. Draxler, R., and G. Hess, 1998: An overview of the HYSPLIT_4 modelling system for trajectories. *Aust. Meteorol. Mag.*, **47**.

3. Wernli, H., and H. C. Davies, 1997: A lagrangian-based analysis of extratropical cyclones. I: The method and some applications. Quart. J. Roy. Meteor. Soc., 123, 467–489

Munir Ahmad Nayak

---

## Referee Comment (RC2) · Anonymous Referee #2 · 1 Jan 2020

**Review of Tuinenburg & Staal (2019)**

**Tracking the global flows of atmospheric moisture**

In this study, a Lagrangian moisture tracking model driven by ERA5 reanalysis data is presented, and recommendations concerning both input data resolution and model set-up in light of accuracy and efficiency are given based on sensitivity experiments. This method is designed for tracking the fate, or as the authors phrase it, the 'footprint' of evaporative moisture released at point locations around the globe. Such a sensitivity analysis comes at the right time, as ERA5 drastically improved both the temporal and spatial resolution as compared to the widely used ERA-Interim in moisture tracking studies, yet this wealth of data is also accompanied with data storage limitations, and considerable simulation time increases. This well-written, nicely presented study may therefore boost scientific progress by providing guidelines on how to use ERA5 data most efficiently, and additionally, demonstrates the power of Lagrangian modelling for tracing moisture. Although generally easy to understand, some descriptions lack clarity, and I believe the manuscript could be ameliorated if the authors considered the following points listed below (not sorted), and thus become suitable for publication in HESS.

- **Number of particles:** If I understand it correctly, then even the setting with the lowest number, 100 particles per mm of evaporation, is still high compared to some previous studies; in case of e.g. Manaus, for a range of say 3 to 6 mm of daily evaporation, this would correspond to 300 to 600 particles being released per day.
Studies such as Gimeno et al. (2012) and many, many others follow the method introduced in Stohl & James (2004) and are based on air parcels (that, altogether, represent the entire atmosphere), and not a 'collection of water particles', as is the case here, but I believe the number of particles may still be compared. Even though some studies based on the Stohl & James (2004) approach do not even mention the number of parcels (or 'particles'), a lot of publications are based on simulations with 6-hourly timesteps for which 2 million air parcels for the entire globe are used (e.g. Sorí et al., 2017; García-Herrera et al., 2019, … ). Läderach & Sodemann (2016) used 5 million parcels globally, which corresponds to an average of about 70 parcels per column residing over each grid cell at any given timestep, so that about 280 parcels are available per grid cell and day, compared to about 120 in many other papers (2 million parcels globally). Therefore, the setting used in this study with 'only' 100 particles per mm of evaporation, at least in the case of high evaporation rates as in Manaus, still results in far more particles than were employed in many peer-reviewed moisture tracking studies. Indeed, as the authors point out in the discussion, larger study areas and longer analysis periods likely decrease the number of particles needed. Still, it would be interesting to see an extended Fig. 4 for e.g. 50, 25 or even 10 particles per mm of evaporation, and this might also reveal a lower limit, since even 'only' 100

particles perform nearly as well as 10'000. Considering the near-linear runtime dependency on the number of particles, this information might be valuable.

- **Particle vs parcel:** Related to the comment above, I am not sure if the usage of 'particle' throughout most of the manuscript is ideal. Particle by definition implies a small size, but depending on the number of particles (per unit of evaporation) used, each particle represents a considerable amount of water, and obviously not a tiny droplet or even a single water molecule, as one might first think. As of now, both 'particle' and 'parcel' are used in the manuscript, so I suggest to remove one of these to ensure consistency, preferably the former.

- **Omega:** It is unclear to me why the authors decided to omit omega in their default configuration. In this context, it is important to point out that the statement on l. 423 is incorrect: the approach of Stohl et al. (2005) does use omega, yet it is complimented by random vertical displacements of air parcels to represent convective (vertical) redistribution. Also, the argument starting on l. 436, "We assume that this speed of mixing is rapid enough to supersede larger-scale vertical flows so as to simplify the model and exclude omega", seems to contradict a previous statement, in which the large effect of omega (vs no omega) is highlighted; moreover, evaporation footprints with and without omega are only similar for (probably unrealistically) fast mixing times of 1 hour, which strongly indicates that this aforementioned 'superseding' does not really take place for mixing at timescales of 24 hours. The lack of a proper reference, since the baseline model is essentially identical to the 3D Lagrangian model (a major limitation that is not really emphasized in the manuscript), clearly makes it difficult to justify any choice here; however, I am not convinced by the explanations provided so far, other than omitting omega 'for simplicity' and to achieve faster simulation times.

- **Horizontal resolution:** Especially because of the authors' valuable recommendation not to degrade the vertical resolution, it would be helpful for the scientific community to know whether the same is true for degrading the horizontal resolution. As stated in the manuscript, many previous Lagrangian studies are based on input data at 0.75x0.75°, or even 1.0x1.0° horizontal resolution, so that upgrading to 0.25x0.25° represents a massive improvement. However, this comes at the cost of extensive input data size, which makes studies covering multiple decades rather than a few days extremely challenging. I suggest adding this analogously to the degradation of vertical information; e.g. for increments of 0.25° from 0.25x0.25° up to 1.0x1.0°. For the same reasons, investigating the sensitivity to temporal resolution would be helpful too, but I am aware that the authors cannot include everything in their manuscript.

- **Tracking time:** 99% of moisture allocated, or 30 days: Dirmeyer & Brubaker (2007) used 90% & 15 days, and nearly all tracking studies do not exceed 15 (or even 10!) days either, as the trajectory accuracy is known to decrease with increasing length (Stohl & Seibert, 1998). Due to how the model is set up, at least if I understand it correctly (see also below), this choice might not really affect the results, but I still suggest to check if the conclusions hold for considerably shorter trajectory lengths, such as 15 or 10 days.

- **Model description:** is insufficient. According to my understanding, tracked moisture remains static until precipitation occurs ('over' the grid cell underneath), and then decreases accordingly to the ratio of precipitation over precipitable water (of the entire column, I presume?). If so, this invokes another assumption, namely that each water (vapor) molecule within a tropospheric column has the same odds of condensing and precipitating (Dirmeyer & Brubaker, 2007). Please clarify.

**Minor comments and suggestions**

In addition to the major comments above, a few additional comments and suggestions are listed here.

- l. 25: This sentence describes *continental* moisture recycling, not the more general concept of moisture recycling as it is most commonly defined (e.g. Brubaker et al., 1993; Dirmeyer et al., 2009). I suggest to include 'Continental' at the beginning of this sentence, and replace 'continental evaporation' e.g. by 'terrestrial evaporation'. This would also be consistent with the results section, where the continental recycling ratio (CRR) is used already.
- l. 113: Not all Lagrangian models are initialized with a collection of water particles, but indeed, this is true for the approach presented in the manuscript. I suggest rephrasing this.
- l. 166: Is this the only instance where the (very short) study period is referred to? I think it would be justified to add a sentence in the discussion to remind the reader of this limitation.
- l. 171: Is there any reason behind the choice of both Utrecht and Stockholm, which are rather 'close' both geographically and climatologically compared to all other point sources, other than the authors' affiliations? This is merely a question/comment, not a suggestion, since the conclusions drawn by the authors do not depend on this choice.
- l. 265: Is the usage of time units (#particle mm$^{-1}$**h$^{-1}$**) correct here? Besides, elsewhere in the text, the number of particles simply relates to some evaporation amount/volume (without referring to time at all), which is already a sufficient description to me.
- l. 306: Do the authors have an explanation as to why the simulation time increases with fewer vertical levels? I do not doubt that this is correct, I simply find it counter-intuitive.
- l. 423 contains an incorrect statement, as Stohl et al., 2005 do not disregard omega in their approach (also mentioned above)
- Fig. 9: Why does B resemble Fig. 3D more than C? According to the text (l. 204), the default choice for the 3D Lagrangian model is 2'000 particles per mm of evaporation.

**Further comments**

Below are a few additional, language-related comments.

- l. 36: "Universal approach**es** and principle**s** moisture tracking models **is** that they apply…", this reads a bit weird to me.
- l. 105: "courser" => coarser?
- l. 127: "weighed" => weighted?

- l. 185: "on full resolution" => at full resolution?
- l. 345: "CRR decreased much more rapidly **with (increasing)** mixing time than ...", maybe a word (or two) went missing here?
- l. 442: "***The*** Continental recycling ratio", not sure if this is correct without any article

**Concluding remarks**

The paper investigates a multitude of choices and assumptions related to setting up a (Lagrangian) moisture tracking framework based on ERA5 data, and even though the main figures are only for a single location (Manaus) among a total of seven, and the analysis time is very short, I believe this choice is justified. Except for the discussion on vertical mixing and omega, as well as the interpretation on the sensitivity to the number of particles, the conclusions stated in the study make sense to me. Additional experiments (e.g. horizontal & temporal information degradation) and some more explanations might complete the package, hence I would gladly serve as a reviewer again and look forward to reading the revised paper.

**References**

Brubaker, K. L., Entekhabi, D. & Eagleson, P. S. Estimation of continental precipitation recycling. *J. Clim.* **6,** 1077–1089 (1993).

Dirmeyer, P. A., Schlosser, C. A. & Brubaker, K. L. Precipitation, recycling, and land memory: An integrated analysis. *J. Hydrometeorol.* **10,** 278–288 (2009).

García-Herrera, R. et al. The European 2016/17 drought. *J. Clim.* **32,** 3169–3187 (2019).

Läderach, A. & Sodemann, H. A revised picture of the atmospheric moisture residence time. *Geophys. Res. Lett.* **43,** 924–933 (2016).

Sodemann, H., Schwierz, C. & Wernli, H. Interannual variability of Greenland winter precipitation sources: Lagrangian moisture diagnostic and North Atlantic Oscillation influence. *J. Geophys. Res.* **113,** 1–17 (2008).

Sorí, R., Nieto, R., Vicente-Serrano, S. M., Drumond, A. & Gimeno, L. A Lagrangian perspective of the hydrological cycle in the Congo River basin. *Earth Syst. Dyn.* **8,** 653–675 (2017).

Stohl, A. & Seibert, P. Accuracy of trajectories as determined from the conservation of meteorological tracers. *Q. J. R. Meteorol. Soc.* **124,** 1465–1484 (1998).

Stohl, A., Forster, C., Frank, A., Seibert, P. & Wotawa, G. Technical note: The Lagrangian particle dispersion model FLEXPART version 6.2. *Atmos. Chem. Phys.* **5,** 2461–2474 (2005).

---

## Author Comment (AC1) · 10 Feb 2020

Reviewer 1
Munir Ahmad Nayak

Dear Dr. Louise

Thank you for sending the manuscript to me for a review. Below you will see a short summary of the manuscript, followed by my specific (major) comments, and then technical corrections (minor comments) at the end.

Summary
Tracking moisture in the atmosphere over time has many applications in the fields of hydrology and meteorology, such as finding the major moisture sources of particular extreme precipitation event at a given location. Moisture tracking models can be represented with a variety of schemes, which include Eulerian and Lagrangian (in two and three dimensions) frameworks, different integration time steps, different sets of vertical forcings, representations of vertical wind velocities, locations of moisture releases to the atmosphere, etc. The results of moisture tracking, for example evaporation recycling rate, the distance travelled by moisture, etc., will depend on the scheme chosen. The authors experiment a set of evaporation tracking schemes to asses the sensitivity of tracking results to different schemes. In summary, the steps used in the manuscript can be written as: a.) Select seven point-sources of evaporation across the globe, b.) track the evaporation (during first five days of July 2012) from these locations using a tracking scheme, c.) keep track of precipitation locations, i. e., latitude and longitude points where precipitation happened, d.) repeat the above three steps with different model settings, and e) compare the tracking results. Based on the comparisons among different schemes, the authors propose an "optimal" tracking scheme for general hydrological applications: 3D Lagrangian, 500 particles released per mm evaporation, moisture releases at surface, linear interpolation in time and space, adding as many vertical forcings as possible, etc.
The manuscript is very clearly written, and overall, I think it can be of interest to many readers of HESS and other similar journals. There, however, are some places in the manuscript where authors should provide more justifications and clarifications; I added these in the "specific comments" section below. I hope the authors can address these comments, after which the manuscript may be suitable for publication in HESS. I will be looking forward to reading a revised draft.

Thank you for the encouragements and we are happy to see that the manuscript was clear.

Specific comments
1. When an air parcel moves, it gains and losses moisture along its track, the gain and loss can be attributed directly to the location where the change happens if the parcel is within the boundary layer. When the parcel is out of the boundary layer, the change locations are not clearly evident, since they can come from remote sources (Sodemann et al., 2008), which are difficult to evaluate. It is not clear how the "original" evaporation (Lines 95-96) is maintained throughout the parcel's track. More clarity on this will help in interpreting many results presented in section 3, for example evaporation footprint.

In the current model, no distinction is made between moisture within the boundary layer and that above it. We are treating the allocation in the same way regardless of the current vertical position of the parcel. That means that we are allocating the surface fluxes (and the moisture gains and losses) to all moisture in the atmospheric column, and thus to all parcels present at that location. The "original evaporation" in this case reflects the evaporation at the location and moment the parcel was released, which is probably a different location than the parcel along the track. We then allocate moisture along the parcel's path according to the ratio of P/PW at the current location of the parcel. As stated, this is done independent of vertical position of the parcel, meaning we assume perfect vertical mixing throughout the atmospheric column. We changed the lines around the former lines 95-96 to state that it is about the amount of original evaporation transported with the parcel (lines 104-108 in the document with tracked changes): "However, once there is precipitation at the location of the parcel, a fraction of the moisture (precipitation over precipitable water of the entire atmospheric column, $\frac{P}{PW}$) that is still present in the parcel is allocated to rain out in that location. This assumes that all moisture in the atmospheric column has the same probability of raining out. Thus, the amount of original evaporation remaining decreases with downwind moisture transport.".

Regarding whether the moisture can be attributed to remote or local sources, we actually think that moisture above the boundary layer can be attributed more easily. If the only difference between the boundary layer (BL) and free troposphere (FT) is the mixing (strong in the BL, not so strong in the FT), the source regions of parcels in the free troposphere can be more easily determined because the parcels will move with the large-scale winds. In the boundary layer, there is more mixing and therefore more vertical displacement of parcels. Given that this study shows that atmospheric moisture transport can be quite sensitive to vertical displacement assumptions, we could assume that it is more difficult to pinpoint the original moisture source of boundary layer moisture compared to free tropospheric moisture. Nevertheless, we did not look at the difference between boundary layer and non-boundary layer moisture here, but consider it worthwhile for future research to test explicitly what the sensitivities of moisture recycling to boundary layer and non-boundary layer vertical mixing are.

2. Section 2.1.4: During convective up- and down drafts, horizontal winds also show significant changes in magnitude and direction; the particles can then be displaced vertical depending on the changes in the vertical winds, instead of assigning random vertical displacements to them, which seems arbitrary. If feasible, another scheme based on this large horizontal wind gradients may be added in the present framework.

Thank you for this suggestion. We agree that the unconditional mixing scheme currently employed is a simplification of reality. As suggested, large-scale conditions may influence the vertical mixing rate. Furthermore, it is possible to use the convective mass fluxes to determine local vertical mixing rates, as we did in Staal et al. (2018, *Nature Climate Change* 8:539-543). As the goal of the current work is to test the sensitivity of different kinds of assumption on moisture tracking, we have chosen to limit the analyses to the current mixing assumptions of four mixing strengths that all happen regardless of atmospheric conditions. For future work, a sensitivity analysis of more physical vertical displacements would be relevant.

3. The basic structure of the model is not presented anywhere. I suggest adding a stepwise

procedure on how the tracking is performed. Actually, response to this might answer my first comment 1 also.

We have rewritten the model description section (2.1) and presented the moisture tracking procedure in a more stepwise way (lines 98-112). Furthermore, we restructured the Methods section with a more logical order of presentation of model structure and assumptions, and experiments.

4. This baseline model is 3D Lagrangian with 10,000 parcels released per mm; the 3D model in L243, table 1, and other results almost identical to the baseline model. This does not seem a reasonable way to compare models and present results, since baseline itself is not "True Tracking" and cannot be a perfect reference. It might be a good idea to use other models' output as reference, such as HYSPLIT (Draxler and Hess, 1998), LAGRANTO (Wernli, H., and H. C. Davies, 1997).

"True tracking" is, of course, impossible. However, a three-dimensional Lagrangian scheme with an extremely large amount of parcels (10,000 for each mm evaporation) uses the available information in the most elaborate way. That is, no information is lost. For this reason, we considered that the best possible baseline. We chose not to compare our results with those form other models in the literature, which were generally developed for ERA-Interim data. Rather, our aim is to test how the ERA5 data can be used best for moisture tracking purposes. Revisiting existing models would address different questions. We prefer to remain within our original scope and not use other models as reference.

5. In Section 3.2, it is argued that number of parcels released does not affect tracking results greatly. We should note, however, that number of parcels may matter to capture convective/converging and diverging events, as stated by the authors in section 2.2.4. Here, the simulations are run only for one case (July 2012), which may not have large convergence or divergence at any time. We should be careful in generalizing these results to all events, unless simulations results of some specific convective events show similar results.

We agree that we must be careful in generalizing our results. Because of possible biases resulting from using only one case, we did our experiments for seven locations spread around the globe. Indeed, some locations were more sensitive to certain changes in model settings than others. However, in general the results were robust. Apart from these seven cases we published our code including options to change the settings according to our methods. Thus, users of the model can perform their own sensitivity analysis if they want to, or change the settings to suit specific research questions. We added in lines 533-534 that we tracked only for one month the moisture released during five days is an additional reason for caution in generalizing our results.

Technical corrections
Define "footprints" at the beginning, somewhere in the introduction.

Thank you for spotting this. We now define "footprint" as "the distribution of precipitation resulting from evaporation from a point or area" in line 50.

One of the aims of the manuscript was to evaluate model structure; however, it is not clear where model structures have changed. Perhaps, Eulerian and Lagranian can be taken as different model structures, but this needs to be written explicitly.

Thank you for spotting this. We now define model structure as "Eulerian or Lagrangian and the number of spatial dimensions" in lines 76-77.

L25-26: Fig. 1 does not specifically show moisture recycling as indicated here.

We removed the reference to Figure 1 in this sentence.

L46-48: Rather than "assumptions", I feel they are more like user "choices".

We agree and rephrased to "choices" in these lines.

L44: Here, I suggest writing "parcels" instead of "particles".

We replaced "particle" with "parcel" throughout the manuscript.

L51-54: It is not clear how the results will be incorrect; also, clearly explain why the Eulerian model simulations will not be as fast as Lagrangian when moisture is released from small areas.

We moved the explanation of why the results will be incorrect from the methods to the introduction in lines 54-56: "If the time step is chosen too large, real moisture transport may occur faster than the simulation grid and time step allow for (i.e., if the Courant number $C = \frac{v\Delta t}{\Delta x} > 1$). If the time step is taken too small, numerical diffusion will occur, meaning that moisture transport in the model will be faster than in the forcing data."

To explain why Eulerian models will not be as fast as Lagrangian for small areas we added "The reason is that they are insensitive to an increase in scale, as all grid cells are updated with the same speed regardless of the amount of moisture present." in lines 58-59.

L60: Do you mean "which resulted in Courant numbers exceeding one …"?

We removed this part of the sentence, because it was unnecessarily complicating.

L125: I am not sure if I understand why vertical mixing is to be carried out every time interval and how is it performed; more details on this can help readers.

The vertical mixing is carried out every time step in the sense that every time step, a random number [0-1] is determined. If this number is smaller than dt/mixing_rate_hour (dt is the internal time step, mixing_rate_hour is the mixing strength in terms of how many times vertical mixing happens on average, in our case 1 hour, 6 hours, 24 hours and 120 hours), the random vertical displacement occurs. By carrying out the mixing procedure (including random number assessment) every time step, two things are achieved:

1. Mixing happens on average once per mixing_rate_hour hours (so once every 1, 6, 24 or 120 hours);
2. The mixing happens at random moments throughout the trajectory, so there is no bias regarding to mixing at specific moments.

We have added some additional description in section 2.1.8 (lines 255-259): "During every time step, there is a small probability (dt/mix-strength) of running the vertical displacement. We summarize these stochastic vertical displacement versions of the model by the mix-strength (unit: hours), or average time for one repositioning of one parcel, which is once per hour, once per six hours, once per 24 hours and once per 120 hours. This procedure ensures that for each parcel, mixing happens on average once in the time period described by the mixing strength and that the mixing happens at random moments during the trajectory. Thus, no biases occur due to mixing at specific prescribed moments.".

L130-L133: Rephrase for more clarity.

We have added units to the data used and remove the "division by the grid cell length" part, which may have been confusing. Now, the section states that the 2D flow speed is driven by the vertically integrated flow speed. Of course the grid cell length is still relevant for the calculation, but as it is not relevant for the 3D vs 2D discussion, we removed it here.

L150: Do you mean "particle" instead of "parcel"? Try to be consistent.

We now consistently refer to parcels.

L153: No, this does not seem realistic; you might not be able to capture convergence or divergence with this scheme, just because it is random.

We agree that it was too strongly phrased. We removed this sentence.

L178: Here, 10,000 particles are released per mm of evaporation over first 5 days of July 2012? Evaporation from a point source at any instant will be transported during each time step; are we releasing parcels at one instant, say t=0, or over multiple time steps (t =0, t=1, and so on.). Add a few lines to clearly explain how parcels are released, and how evaporation over 5 days will be captured by parcels released.

Yes, 10,000 parcels are released for every mm of evaporation during those five days. We added "All evaporation within this period is accounted for" in lines 164-165.

L189: In table 1: I would also add a simple metric "mean distance travelled".

Thank you for this suggestion. When we ran our simulations, unfortunately, we did not record the distance that the parcels travelled through the atmosphere. We did, however, record the distances between the source locations and those of the sink locations. This is already present in Table 1 (mean latitudinal distance and mean longitudinal distance).

L225-234: The entire section can be as a separate row in Table 1.

Thank you for this suggestion. We added new rows to Table 1 with the results presented in section 3.1, which are averages of the data that were already provided.

L250: Low value of CRR is observed in 2D Lagrangian case, not Eulerian; see Figure 3c also

Thank you for spotting this. We meant the high value of CRR in the 2D Eulerian case and corrected the sentence accordingly.

L254: Fig. S5, not S4.

Thank you for spotting this. We corrected it.

L259-262: Give clear reasons of so much computational time difference between 3D Lagrangian and 3D Eulerian schemes. In other words, why do we think 3-D Eulerian takes so much time?

The reason is that Eulerian models update the moisture content of all grid cells, even when the actual tracked moisture is in just a part of it. This information is added in lines 58-59.

Figures 6, 7 and other similar figures: Since these figures do not shown clear differences, perhaps it is better to show differences directly, i.e., map of baseline footprint minus footprint from the given set up.

We appreciate this suggestion, but showing differences would serve another purpose than ours. Lack of difference among simulations is—in our case—a relevant result. If we chose to display the differences (only) in the cases where they are very small, the collection of figures would highlight the model settings to which the model is insensitive to rather than the ones that it is sensitive to.

Figure S25B: There is an unusual straight color line in this panel; can that be removed?

Thank you for spotting this. This straight line was due to an error in the plotting. The ERA5 data (and our model) runs on longitudes from 0-360, while Matplotlib expects longitudes from -180-180. We made an error shifting grids, which we now corrected.

Section 3.5: Were the models run with interpolation or with nearest neighbor method. Also, the results here can be concisely presented in a tabular form, rather than text.

As explained in section 2.2.4, all Lagrangian runs were forced by interpolated data, unless stated otherwise, in which case the nearest neighbor was used. Thus, mentioned section uses interpolation.

Section 3.6: Will it be feasible to test sensitivity to timestep dt = 3 hours?

Yes, it is. We expanded our sensitivity analysis to time steps, now including dt = 3h and dt = 6h. The respective figures in the main text and the supplement were updated with two new panels. Also the Methods (section 2.2.7) and the Results (section 3.7) were updated accordingly. Note that time steps >1h imply a degraded temporal resolution compared to the forcing data. We added this information in lines 233-234: "… 3 h, and 6 h. Note that the latter two imply a degradation of the temporal resolution. For these cases we averaged hourly data on wind speed and direction." The additional tests did not change the settings of our optimal model.

Section 3.7: I am not sure if I clearly understand the purpose of mixing and its usefulness in practice. Perhaps provide more details.

The purpose of the vertical mixing is to approximate the role of turbulence in atmospheric moisture transport. Turbulence may be a very important driver of mixing, but this is not covered by reanalysis data. Accounting for vertical mixing as we do may compensate for that. We clarified this in lines 388-389: "Turbulence may cause considerable vertical mixing in the atmosphere, but because the rate of this mixing is unknown, …".

References
1. Sodemann, H., C. Schwierz, and H. Wernli, 2008: Interannual variability of Greenland winter precipitation sources: Lagrangian moisture diagnostic and North Atlantic Oscillation influence. *J. Geophys. Res.*, **113**, D03107.
2. R. Draxler, R., and G. Hess, 1998: An overview of the HYSPLIT_4 modelling system for trajectories. *Aust. Meteorol. Mag.*, **47**.
3. Wernli, H., and H. C. Davies, 1997: A lagrangian-based analysis of extratropical cyclones. I: The method and some applications. Quart. J. Roy. Meteor. Soc., 123, 467–

---

## Author Comment (AC2) · 10 Feb 2020

Reviewer 2

In this study, a Lagrangian moisture tracking model driven by ERA5 reanalysis data is presented, and recommendations concerning both input data resolution and model set-up in light of accuracy and efficiency are given based on sensitivity experiments. This method is designed for tracking the fate, or as the authors phrase it, the 'footprint' of evaporative moisture released at point locations around the globe. Such a sensitivity analysis comes at the right time, as ERA5 drastically improved both the temporal and spatial resolution as compared to the widely used ERA-Interim in moisture tracking studies, yet this wealth of data is also accompanied with data storage limitations, and considerable simulation time increases. This well-written, nicely presented study may therefore boost scientific progress by providing guidelines on how to use ERA5 data most efficiently, and additionally, demonstrates the power of Lagrangian modelling for tracing moisture. Although generally easy to understand, some descriptions lack clarity, and I believe the manuscript could be ameliorated if the authors considered the following points listed below (not sorted), and thus become suitable for publication in HESS.

Thank you for these encouraging words.

- **Number of particles:** If I understand it correctly, then even the setting with the lowest number, 100 particles per mm of evaporation, is still high compared to some previous studies; in case of e.g. Manaus, for a range of say 3 to 6 mm of daily evaporation, this would correspond to 300 to 600 particles being released per day.
Studies such as Gimeno et al. (2012) and many, many others follow the method introduced in Stohl & James (2004) and are based on air parcels (that, altogether, represent the entire atmosphere), and not a 'collection of water particles', as is the case here, but I believe the number of particles may still be compared. Even though some studies based on the Stohl & James (2004) approach do not even mention the number of parcels (or 'particles'), a lot of publications are based on simulations with 6-hourly timesteps for which 2 million air parcels for the entire globe are used (e.g. Sorí et al., 2017; García-Herrera et al., 2019, … ). Läderach & Sodemann (2016) used 5 million parcels globally, which corresponds to an average of about 70 parcels per column residing over each grid cell at any given timestep, so that about 280 parcels are available per grid cell and day, compared to about 120 in many other papers (2 million parcels globally). Therefore, the setting used in this study with 'only' 100 particles per mm of evaporation, at least in the case of high evaporation rates as in Manaus, still results in far more particles than were employed in many peer-reviewed moisture tracking studies. Indeed, as the authors point out in the discussion, larger study areas and longer analysis periods likely decrease the number of particles needed. Still, it would be interesting to see an extended Fig. 4 for e.g. 50, 25 or even 10 particles per mm of evaporation, and this might also reveal a lower limit, since even 'only' 100 particles perform nearly as well as 10'000. Considering the near-linear runtime dependency on the number of particles, this information might be valuable.

Thank you for this suggestion and we agree. Therefore, we extended our sensitivity analysis for the number of parcels with 50 and 10 parcels per mm evaporation. We added two new panels to

the respective figures in the main text (Fig. 4) and the supplement (Figs. S8-S13), and updated the respective sections in the Methods (2.2.2) and Results (3.2). Lowering the amount of parcels in this way had a small effect on the statistics and made the footprints less smooth. Therefore, we decided to keep the setting of our optimal model at 500 parcels mm$^{-1}$ evaporation. Moreover, we mentioned in the methods section that the number of parcels used here is high compared to other moisture tracking studies (lines 194-196 in the version with tracked changes).

- **Particle vs parcel:** Related to the comment above, I am not sure if the usage of 'particle' throughout most of the manuscript is ideal. Particle by definition implies a small size, but depending on the number of particles (per unit of evaporation) used, each particle represents a considerable amount of water, and obviously not a tiny droplet or even a single water molecule, as one might first think. As of now, both 'particle' and 'parcel' are used in the manuscript, so I suggest to remove one of these to ensure consistency, preferably the former.

We now consistently refer to parcels throughout the manuscript.

- **Omega:** It is unclear to me why the authors decided to omit omega in their default configuration. In this context, it is important to point out that the statement on l. 423 is incorrect: the approach of Stohl et al. (2005) does use omega, yet it is complimented by random vertical displacements of air parcels to represent convective (vertical) redistribution.

Thanks, we changed the statement to reflect that the study by Stohl et al. (2005) does include omega and uses a scheme based on turbulent mixing to create additional vertical displacements (lines 487-489): "Therefore, moisture tracking models may complement the omega-based vertical displacement by using a mixing scheme based on turbulent mixing (Stohl et al., 2005), or …".

Also, the argument starting on l. 436, "We assume that this speed of mixing is rapid enough to supersede larger-scale vertical flows so as to simplify the model and exclude omega", seems to contradict a previous statement, in which the large effect of omega (vs no omega) is highlighted; moreover, evaporation footprints with and without omega are only similar for (probably unrealistically) fast mixing times of 1 hour, which strongly indicates that this aforementioned 'superseding' does not really take place for mixing at timescales of 24 hours. The lack of a proper reference, since the baseline model is essentially identical to the 3D Lagrangian model (a major limitation that is not really emphasized in the manuscript), clearly makes it difficult to justify any choice here; however, I am not convinced by the explanations provided so far, other than omitting omega 'for simplicity' and to achieve faster simulation times.

Thanks for raising this issue. We agree that we may have been too quick to disregard inclusion of omega for the weaker mixing assumptions. It is true that the baseline model is 'just' the 3D Lagrangian model, because it is quite hard to get reliable mixing, or source-sink relations of atmospheric moisture, especially in the context of the current study in which many sensitivity test are performed.

Nevertheless, we changed the manuscript in the method section to state that there are some sensitivity analyses for which we have a true value of moisture recycling (the numerical ones; time step, degraded resolutions, number of particles, etc.) and there are some for which we do not have a true estimate (vertical mixing, release height) (lines 173-176): "For some of the sensitivity tests, these criteria are evaluated against the simulation with the most detailed settings (most parcels, highest resolution, etc.), in which case there is a numerical true estimate. However, for some tests, there is no information to derive a true value. For these tests, the uncertainty remains higher and we derive the sensitivity of moisture recycling to the assumptions."

We have changed the discussion (lines 502-505) and conclusion (lines 547-548) to reflect that the vertical transport is the main uncertainty regarding moisture tracking.

As a small note, given the data volume of the ERA5 data, the reason for excluding the omega data (or reducing the forcing data in general) has more to do with computer memory (RAM) use of the simulation than with the CPU time needed. Once all data is loaded into RAM, looking up specific values is done in about constant time.

- **Horizontal resolution:** Especially because of the authors' valuable recommendation not to degrade the vertical resolution, it would be helpful for the scientific community to know whether the same is true for degrading the horizontal resolution. As stated in the manuscript, many previous Lagrangian studies are based on input data at 0.75x0.75°, or even 1.0x1.0° horizontal resolution, so that upgrading to 0.25x0.25° represents a massive improvement. However, this comes at the cost of extensive input data size, which makes studies covering multiple decades rather than a few days extremely challenging. I suggest adding this analogously to the degradation of vertical information; e.g. for increments of 0.25° from 0.25x0.25° up to 1.0x1.0°. For the same reasons, investigating the sensitivity to temporal resolution would be helpful too, but I am aware that the authors cannot include everything in their manuscript.

Thank you for this suggestion and we agree that testing for degrading horizontal resolution is a worthwhile addition to our paper. Therefore, apart from 0.25°, we performed our analyses also for 0.5°, 1.0°, and 1.5°. We added a new Fig. 8, supplementary Figs. S32-37 and new sections 2.2.6 and 3.6.

The horizontal degradation had quite considerable effects on the statistics, so we decided not to change the settings of our optimal model based on these new experiments.

- **Tracking time:** 99% of moisture allocated, or 30 days: Dirmeyer & Brubaker (2007) used 90% & 15 days, and nearly all tracking studies do not exceed 15 (or even 10!) days either, as the trajectory accuracy is known to decrease with increasing length (Stohl & Seibert, 1998). Due to how the model is set up, at least if I understand it correctly (see also below), this choice might not really affect the results, but I still suggest to check if the conclusions hold for considerably shorter trajectory lengths, such as 15 or 10 days.

We appreciate the suggestion, but we would advise the model users against using short tracking times. It happens often that after ten days not all tracked moisture has been allocated yet, in which case continuing the tracking will be better than terminating it. This happens especially over drier areas. Although we agree that the uncertainty increases with time, the quality of the forcing data is not reduced with longer tracking, as they are observation-based and not model-based.

- **Model description:** is insufficient. According to my understanding, tracked moisture remains static until precipitation occurs ('over' the grid cell underneath), and then decreases accordingly to the ratio of precipitation over precipitable water (of the entire column, I presume?). If so, this invokes another assumption, namely that each water (vapor) molecule within a tropospheric column has the same odds of condensing and precipitating (Dirmeyer & Brubaker, 2007). Please clarify.

Yes, this is correct. The allocation happens according to the ratio of P/PW, in which PW is the precipitable water of the entire column. This indeed assumes that every unit of water in the entire column has an equal probability of precipitating out of the column. We have added this fact and assumption explicitly in the model description section (lines 104-107): "However, once there is precipitation, a fraction of the moisture (precipitation over precipitable water of the entire atmospheric column, $\frac{P}{PW}$) that is still present in the atmosphere is allocated to rain out in that location. This assumes that all moisture in the atmospheric column has the same probability of raining out."

Minor comments and suggestions
In addition to the major comments above, a few additional comments and suggestions are listed here.
- l. 25: This sentence describes *continental* moisture recycling, not the more general concept of moisture recycling as it is most commonly defined (e.g. Brubaker et al., 1993; Dirmeyer et al., 2009). I suggest to include 'Continental' at the beginning of this sentence, and replace 'continental evaporation' e.g. by 'terrestrial evaporation'. This would also be consistent with the results section, where the continental recycling ratio (CRR) is used already.

Thank you, we revised it as suggested.

- l. 113: Not all Lagrangian models are initialized with a collection of water particles, but indeed, this is true for the approach presented in the manuscript. I suggest rephrasing this.

We rephrased this sentence to "In Lagrangian models, the internal model state is not a model grid, but generally a collection of water parcels." (line 128).

- l. 166: Is this the only instance where the (very short) study period is referred to? I think it would be justified to add a sentence in the discussion to remind the reader of this limitation.

In lines 533-534 in the Discussion, we added "… that we tracked only moisture for one month that evaporated during five days mean[s] that one should be cautious with generalizing the implications of these outcomes."

- l. 171: Is there any reason behind the choice of both Utrecht and Stockholm, which are rather 'close' both geographically and climatologically compared to all other point sources, other than the authors' affiliations? This is merely a question/comment, not a suggestion, since the conclusions drawn by the authors do not depend on this choice.

As you suspected, the choice of these two European study locations is no coincidence. Indeed, the two are relatively close, but we found it worthwhile to include a high-latitude evaporation source in addition to a more centrally located European location, as moisture recycling models have not always been performing well at higher latitudes. But apart from that, choosing both Utrecht and Stockholm helps us to communicate (the performance of) our model at our institutions.

- l. 265: Is the usage of time units (#particle mm -1 **h -1** ) correct here? Besides, elsewhere in the text, the number of particles simply relates to some evaporation amount/volume (without referring to time at all), which is already a sufficient description to me.

Thank you for spotting this mistake. We deleted the reference to time in this sentence.

- l. 306: Do the authors have an explanation as to why the simulation time increases with fewer vertical levels? I do not doubt that this is correct, I simply find it counter-intuitive.

We agree that this is counter intuitive, as all simulations were run on the same computer. However, the simulations for the degraded vertical were run at a different moment than the other ones, which may explain the small difference of 1% in their CPU time.

- l. 423 contains an incorrect statement, as Stohl et al., 2005 do not disregard omega in their approach (also mentioned above)

We apologize for this incorrect reference and modified the example it was based on.

- Fig. 9: Why does B resemble Fig. 3D more than C? According to the text (l. 204), the default choice for the 3D Lagrangian model is 2'000 particles per mm of evaporation.

Figure 9B (now Figure 10B) equals Figure 3D, because both use the default settings of the model: no omega, 6h mixing, and 2000 parcels. All panels in Figure 10 use 2000 parcels per mm.

Further comments
Below are a few additional, language-related comments.
- l. 36: "Universal approach **es** and principle **s** moisture tracking models **is** that they

apply…", this reads a bit weird to me.
- l. 105: "courser" => coarser?
- l. 127: "weighed" => weighted?
3
- l. 185: "on full resolution" => at full resolution?
- l. 345: "CRR decreased much more rapidly **with (increasing)** mixing time than ...",
maybe a word (or two) went missing here?
- l. 442: " **The** Continental recycling ratio", not sure if this is correct without any article

Thank you very much for these detailed comments. We fixed these mistakes. In the last case we
changed it to "Continental recycling ratios …".

Concluding remarks
The paper investigates a multitude of choices and assumptions related to setting up a
(Lagrangian) moisture tracking framework based on ERA5 data, and even though the main
figures are only for a single location (Manaus) among a total of seven, and the analysis time is
very short, I believe this choice is justified. Except for the discussion on vertical mixing and
omega, as well as the interpretation on the sensitivity to the number of particles, the
conclusions stated in the study make sense to me. Additional experiments (e.g. horizontal &
temporal information degradation) and some more explanations might complete the package,
hence I would gladly serve as a reviewer again and look forward to reading the revised paper.

Thank you very much.

References
Brubaker, K. L., Entekhabi, D. & Eagleson, P. S. Estimation of continental precipitation
recycling. *J. Clim.* **6,** 1077–1089 (1993).
Dirmeyer, P. A., Schlosser, C. A. & Brubaker, K. L. Precipitation, recycling, and land memory:
An integrated analysis. *J. Hydrometeorol.* **10,** 278–288 (2009).
García-Herrera, R. et al. The European 2016/17 drought. *J. Clim.* **32,** 3169–3187 (2019).
Läderach, A. & Sodemann, H. A revised picture of the atmospheric moisture residence time.
*Geophys. Res. Lett.* **43,** 924–933 (2016).
Sodemann, H., Schwierz, C. & Wernli, H. Interannual variability of Greenland winter
precipitation sources: Lagrangian moisture diagnostic and North Atlantic Oscillation influence.
*J. Geophys. Res.* **113,** 1–17 (2008).
Sorí, R., Nieto, R., Vicente-Serrano, S. M., Drumond, A. & Gimeno, L. A Lagrangian
perspective
of the hydrological cycle in the Congo River basin. *Earth Syst. Dyn.* **8,** 653–675 (2017).
Stohl, A. & Seibert, P. Accuracy of trajectories as determined from the conservation of
meteorological tracers. *Q. J. R. Meteorol. Soc.* **124,** 1465–1484 (1998).
Stohl, A., Forster, C., Frank, A., Seibert, P. & Wotawa, G. Technical note: The Lagrangian
particle dispersion model FLEXPART version 6.2. *Atmos. Chem. Phys.* **5,** 2461–2474 (2005).

---

## Referee Report (RR1)

Review of hess-2019-597 "Tracking the global flows of atmospheric moisture" Obbe A. Tuinenburg and Arie Staal

Dear Dr. Louise,

Thank you for sending the manuscript for a re-review. I read the responses to my reviews and the corresponding changes made in the revised manuscript. The authors have addressed most of my minor technical comments and to some extent my specific major comments. However, I feel that a few of my specific comments should to be addressed more scientifically.

Specific Comments:

1. Regarding my comment 4: Surely "True tracking" is not possible, but I strongly feel that an independent tracking model should be used to generate observations for a reasonable verification and comparison of different settings here. Given that the present operational moisture tracking models, such as Hyspilt", are not hard to run and that only one month (July 2012) is to be evaluated, I think adding this independent set of observations would be a valuable addition to the manuscript.

2. In response to my specific comment 3, the structure added in methods section seems to focus on Lagrangian trajectory analysis only; the authors may modify it to incorporate Eulerian framework as well.

3. In the section 3.2, "The number of parcels has a small effect on the level of detail" on line 290, a cautionary note should be added for convective and other similar events, where the number of particles may play a significant role.

4. The title of the manuscript is too broad, perhaps it can be made more specific to sensitivity analysis of different tracking choices.

Minor:

1. Line 155-166. Please rephrase the two sentences for more clarity.

Munir Ahmad Nayak

2. Section 3.1 Paragraph 1: Again, the numbers are already in Table 1 for the reader to look at quantities. In this section, you might want to present general clear qualitative differences, just in one or two lines. The main text should not be overloaded with numbers and mathematical terms, see for example Lines 360 to 370. It would be much easier to read qualitative information than numbers, and the amount of text would be reduced significantly. For example, the Lines 369-370 can be reduced to one line by saying "increasing the temporal resolution increased the running time in all cases."

3. Figures 6, 7 and other similar figures: Since the results are on difference, for example L325, I suggest showing relative differences for a clear visual and quantitative evaluation.

4. Figure S24B: The unusual straight color line in this panel should be removed.

Munir Ahmad Nayak

---

## Referee Report (RR2)

**Review II of Tuinenburg & Staal (2019)**

**Tracking the global flows of atmospheric moisture**

I am glad that the authors have extended their analysis and parts of the manuscript, and believe this has improved the quality of the latter. I thus thank the authors for taking my comments into account. In general, I agree with the responses to my review, and believe the manuscript is now suitable for publication – and would be even more so if my (minor) comment below were considered. I have also made a few additional comments and suggestions on the updated manuscript, listed under "Further comments".

**Minor comment**
"- Tracking time: 99% of moisture allocated, or 30 days: Dirmeyer & Brubaker (2007) used 90% & 15 days, and nearly all tracking studies do not exceed 15 (or even 10!) days either, as the trajectory accuracy is known to decrease with increasing length (Stohl & Seibert, 1998). Due to how the model is set up, at least if I understand it correctly (see also below), this choice might not really affect the results, but I still suggest to check if the conclusions hold for considerably shorter trajectory lengths, such as 15 or 10 days."

> "We appreciate the suggestion, but we would advise the model users against using short tracking times. It happens often that after ten days not all tracked moisture has been allocated yet, in which case continuing the tracking will be better than terminating it. This happens especially over drier areas. Although we agree that the uncertainty increases with time, the quality of the forcing data is not reduced with longer tracking, as they are observation-based and not model-based."

With the approach used and described here, evaporation, represented by water vapor parcels, is tracked until less than 1% of the original amount remains, or 30 days are exceeded. Since each time a precipitation event occurs, the water content of parcels is depleted in accordance with the entire column (as now also described more clearly in the Methods), this means that the amount of allocated moisture increases logarithmically – the first few precipitation events (with respect to the beginning of tracking) naturally "remove" larger amounts of the originally evaporated and tracked moisture than after, e.g., 15 days, when a lot of tracked moisture has usually already rained out. Consequently, while extending trajectories from 5 to 10 days has a strong impact on the moisture sink (or source, depending on the approach) region, differences between 15 and 20 days are already marginal (Sodemann & Stohl, 2009). Whereas I would still exert caution in the interpretation of moisture sink regions that were obtained with trajectory lengths up to one month, it is indeed, as the authors point out, the only way to attribute nearly all of the tracked moisture. Personally, I would

include an, e.g., 95% option, possibly in combination with a maximum allowed trajectory length of e.g. 15 days, so that the user can easily choose between
a.) 99%, default: maximizing amount of 'explained' moisture
b.) 95%: maximizing efficiency* & accuracy (of trajectories and hence simulated 'fate of evaporation', or downwind precipitation)

\* I suspect if tracking was always halted at 95%, trajectories would be frequently cut off far 'sooner' than after 30 days, and thus cut down runtime.

However, because the vast majority of moisture reductions (in terms of amount) tends to occur within the first few days (Sodemann, 2020), and even though the authors argued against lowering their maximum allowed trajectory length of 30 days without exploring further, I believe their choice does not affect any of the conclusions drawn in the study. Moreover, unfortunately, while it is clear that trajectory uncertainty does increase with tracking time, assessing this in detail would probably require a publication of its own. Thus, while I believe that the trajectory accuracy constraints I mentioned remain valid, as even the (high-)quality forcing data cannot prevent the accumulation of trajectory errors, I would like to encourage the authors to at least (briefly) inform the reader about this uncertainty, but do not see the need to insist on any additional changes.

**Further comments**
- p. 2, l. 54:
  "If the time step is chosen too large, real moisture transport may occur faster than the simulation grid and time step allow for (i.e., if the Courant number $C = v\Delta t/\Delta x > 1$). If the time step is taken too small, numerical diffusion will occur, meaning that moisture transport in the model will be faster than in the forcing data."

  I am no expert in numerical modelling, but I believe that this phrasing is a bit misleading. As far as I know, numerical diffusion is primarily a consequence of the use of numerical schemes to represent partial differential equations as finite differences, which enables a numerical rather than analytical (and thus exact) solution. While particularly higher-order schemes provide very good approximations, even those are still associated with tiny errors, which, together with discretization errors and machine precision limitations, cause 'numerical diffusion'. I thus suggest to slightly rephrase the second quoted sentence to reflect that numerical diffusion becomes problematic for small timesteps (rather than implying the former is solely caused by the latter), because of course, if there are inaccuracies in the calculation of e.g. wind speeds due to the use of some advection scheme, this becomes exacerbated if these calculations are repeated more often (due to smaller timesteps).

- p. 4, l. 94:
  "In general, atmospheric moisture tracking is achieved by [...] keeping track of how much of that moisture rains out **where**."

  To me, it reads better if the 'where' is moved to '... keeping track **where** and how much of that moisture rains out'.

- p. 13, l. 411:
  "Also here, the figures show that slower vertical mixing increases the area where rainfall depends on evaporation from the studied sources. However, with omega, the rainfall from the sources is more equally distributed within the footprints than without omega. In other words, with omega the footprints show a pattern that is less influenced by diffusion (Figs. 10, S44−S49)."

  The first and second sentences make perfect sense to me, but not the third one. I suppose it is true that there are stronger contrasts in the top row (no omega) of e.g. Fig. 10 as compared to the bottom row (omega), or phrased differently, rainfall is more equally distributed with (than without) omega. But then, considering how the footprints with omega are more smooth, and diffusion (by definition) diminishes gradients, how are the omega-footprints less influenced by diffusion?

- p. 14, l. 444:
  "Also, the distances of moisture transport differed by several degrees for each of these models."

  I believe a comma after 'Also' would help here.

- P. 15, l. 467:
  " … , surface release will be the default in our model."

  Perhaps "... , surface release **is** the default in our model" instead? I believe this would also be consistent with sentences further below, e.g. "As default we take a mixing time of 24 h without omega."

**References**

Sodemann, H. & Stohl, A. Asymmetries in the moisture origin of Antarctic precipitation. *Geophys. Res. Lett.* **36**, 1–5 (2009).

Sodemann, H. Beyond Turnover Time : Constraining the Lifetime Distribution of Water Vapor from Simple and Complex Approaches. *J. Atmos. Sci.* **77,** 413–433 (2020).

---

## Author Response (AR2)

Dear Editor,

Thank you for your rapid decision despite these unusual times, and for giving us a minor revision for our manuscript "Tracking the global flows of atmospheric moisture". We thank the reviewers for their careful and constructive reviews. This has led to a further improvement of the paper, although we have implemented not all suggestions. Please find below the reviews with our responses in blue. Where we refer to line numbers, they apply to the manuscript version with tracked changes.

We hope that after these minor revisions, our manuscript is now deemed suitable for publication.

Please let us know if you have any further questions or requests.

Kind regards,

Obbe Tuinenburg and Arie Staal

Reviewer 1

Munir Ahmad Nayak
Review of hess-2019-597 "Tracking the global flows of atmospheric moisture" Obbe A. Tuinenburg and Arie Staal

Dear Dr. Louise,

Thank you for sending the manuscript for a re-review. I read the responses to my reviews and the corresponding changes made in the revised manuscript. The authors have addressed most of my minor technical comments and to some extent my specific major comments. However, I feel that a few of my specific comments should to be addressed more scientifically.

Thank you. We are glad to read you appreciated the requested revisions, including additional analyses. We hope that our latest revisions are satisfactory as well.

Specific Comments:
1. Regarding my comment 4: Surely "True tracking" is not possible, but I strongly feel that an independent tracking model should be used to generate observations for a reasonable verification and comparison of different settings here. Given that the present operational moisture tracking models, such as Hyspilt", are not hard to run and that only one month (July 2012) is to be evaluated, I think adding this independent set of observations would be a valuable addition to the manuscript.

Thank you for this suggestion. In general, we fully agree with the value of comparing multiple models. Indeed, the purpose of our manuscript is to provide such a comparison. We chose to conduct a systematic sensitivity analysis given the latest ERA5 data. No model in the literature has yet been developed for these data. Although it is possible to compare our results to those of other models, including Hyspilt but also a number of other ones, we strongly argue that this is outside the scope of our work. These models have a number of different assumptions (and data input) and any comparison would fall outside our study set-up. Instead, we encourage the moisture tracking

community to compare and/or improve their models based on our sensitivity analyses and recommendations. This is one of the reasons why we make our model code publicly available.

2. In response to my specific comment 3, the structure added in methods section seems to focus on Lagrangian trajectory analysis only; the authors may modify it to incorporate Eulerian framework as well.

We can see why this appears to be specific for the Lagrangian analysis only, as we refer to parcels. However, the procedure described here applies to Lagrangian and Eulerian. We revised the text in the following way (lines 96-100):

"The following stepwise procedure is employed to do the moisture tracking. At the starting location, a given amount of moisture enters the atmosphere through evaporation. This is the original amount of moisture to be tracked. In a Eulerian setting, this is done on a per-grid-cell basis; in a Lagrangian setting, we track individual units that we call parcels, which is the terminology we use to describe the procedure."

3. In the section 3.2, "The number of parcels has a small effect on the level of detail" on line 290, a cautionary note should be added for convective and other similar events, where the number of particles may play a significant role.

We revised the text (lines 290-291) to "The number of parcels has a small effect on the level of detail in our case studies (although this may be different for convective events)."

4. The title of the manuscript is too broad, perhaps it can be made more specific to sensitivity analysis of different tracking choices.

Thank you for this suggestion. We have considered a more specific title in the earlier stages of the process, but decided against it. We believe that our title reflects both the methodological nature of our work as well as its global scope. We also believe this work is of relevance to everyone working on or with atmospheric moisture tracking; we hope that a general title attracts the attention of the (wider) moisture recycling community.

Minor:
1. Line 155-166. Please rephrase the two sentences for more clarity.

We assume you refer to the two sentences in lines 155-156. We rephrased them as follows (lines 156-157): "We track all moisture that has evaporated from seven source locations during the first five days of July 2012."

2. Section 3.1 Paragraph 1: Again, the numbers are already in Table 1 for the reader to look at quantities. In this section, you might want to present general clear qualitative differences, just in one or two lines. The main text should not be overloaded with numbers and mathematical terms, see for example Lines 360 to 370. It would be much easier to read qualitative information than numbers, and the amount of text would be reduced significantly. For example, the Lines 369-370 can be reduced to one line by saying "increasing the temporal resolution increased the running time in all cases."

Indeed, the individual numbers for this section are also included in Table 1. Not all our results are included in a table, though, which is why we cannot revert to presenting qualitative results only. We consider the table an additional reference for the Eulerian-Lagrangian and 2D-3D comparisons,

because this is relatively important. We maintain the structure of the Results section consistent throughout the manuscript, which is why we not simply refer to Table 1 for the results in Section 3.1.

We added the suggested sentence in line 371: "Increasing the temporal resolution increase the running time in all cases."

3. Figures 6, 7 and other similar figures: Since the results are on difference, for example L325, I suggest showing relative differences for a clear visual and quantitative evaluation.

Thank you for this suggestion. However, we already argued in our response to your previous review: "We appreciate this suggestion, but showing differences would serve another purpose than ours. Lack of difference among simulations is—in our case—a relevant result. If we chose to display the differences (only) in the cases where they are very small, the collection of figures would highlight the model settings to which the model is insensitive to rather than the ones that it is sensitive to."

We see no new reasons to change our minds in this matter.

4. Figure S24B: The unusual straight color line in this panel should be removed.

Thank you very much for your careful look. We thought we had removed all errors in the zero-meridian in the previous round, but apparently we have missed this one. We replaced Figure S24, which is now correct.

Reviewer 2

Review II of Tuinenburg & Staal (2019)

Tracking the global flows of atmospheric moisture

I am glad that the authors have extended their analysis and parts of the manuscript, and believe this has improved the quality of the latter. I thus thank the authors for taking my comments into account. In general, I agree with the responses to my review, and believe the manuscript is now suitable for publication – and would be even more so if my (minor) comment below were considered. I have also made a few additional comments and suggestions on the updated manuscript, listed under "Further comments".

Thank you and we are happy that you consider our work suitable for publication. See below our response to your last comments.

**Minor comment**
"- Tracking time: 99% of moisture allocated, or 30 days: Dirmeyer & Brubaker (2007) used 90% & 15 days, and nearly all tracking studies do not exceed 15 (or even 10!) days either, as the trajectory accuracy is known to decrease with increasing length (Stohl & Seibert, 1998). Due to how the model is set up, at least if I understand it correctly (see also below), this choice might not really affect the results, but I still suggest to check if the conclusions hold for considerably shorter trajectory lengths, such as 15 or 10 days."

"We appreciate the suggestion, but we would advise the model users against using short tracking times. It happens often that after ten days not all tracked moisture has been allocated yet, in which case continuing the tracking will be better than terminating

it. This happens especially over drier areas. Although we agree that the uncertainty increases with time, the quality of the forcing data is not reduced with longer tracking, as they are observation-based and not model-based."

With the approach used and described here, evaporation, represented by water vapor parcels, is tracked until less than 1% of the original amount remains, or 30 days are exceeded. Since each time a precipitation event occurs, the water content of parcels is depleted in accordance with the entire column (as now also described more clearly in the Methods), this means that the amount of allocated moisture increases logarithmically – the first few precipitation events (with respect to the beginning of tracking) naturally "remove" larger amounts of the originally evaporated and tracked moisture than after, e.g., 15 days, when a lot of tracked moisture has usually already rained out. Consequently, while extending trajectories from 5 to 10 days has a strong impact on the moisture sink (or source, depending on the approach) region, differences between 15 and 20 days are already marginal (Sodemann & Stohl, 2009).
Whereas I would still exert caution in the interpretation of moisture sink regions that were obtained with trajectory lengths up to one month, it is indeed, as the authors point out, the only way to attribute nearly all of the tracked moisture. Personally, I would include an, e.g., 95% option, possibly in combination with a maximum allowed trajectory length of e.g. 15 days, so that the user can easily choose between
a.) 99%, default: maximizing amount of 'explained' moisture
b.) 95%: maximizing efficiency* & accuracy (of trajectories and hence simulated 'fate of evaporation', or downwind precipitation)
* I suspect if tracking was always halted at 95%, trajectories would be frequently cut off far 'sooner' than after 30 days, and thus cut down runtime.
However, because the vast majority of moisture reductions (in terms of amount) tends to occur within the first few days (Sodemann, 2020), and even though the authors argued against lowering their maximum allowed trajectory length of 30 days without exploring further, I believe their choice does not affect any of the conclusions drawn in the study. Moreover, unfortunately, while it is clear that trajectory uncertainty does increase with tracking time, assessing this in detail would probably require a publication of its own. Thus, while I believe that the trajectory accuracy constraints I mentioned remain valid, as even the (high-)quality forcing data cannot prevent the accumulation of trajectory errors, I would like to encourage the authors to at least (briefly) inform the reader about this uncertainty, but do not see the need to insist on any additional changes.

Thank you for this explanation and for recognizing the sufficiency of a caution rather than redoing or expanding the results with a round of simulations. We believe the suggestion of some cautionary words about this is a good one. We added the following text in the Discussion (lines 517-525):

"In this study, we track the moisture for up to 30 days. It should be noted that the accuracy of moisture tracking results decreases with tracking time (Sodemann & Stohl, 2009). This is due to the fact that moisture convergence and divergence patterns in the forcing dataset are well represented if there are many parcels present around that location. However, many days after the release of the parcel, it will be transported far away from the release location. Therefore, the density of the parcels released from a specific location will decrease with time and also decrease the moisture tracking accuracy. This is the reason many moisture tracking models stop their simulations after ten days. We chose not to do this, because although in many cases almost all the moisture has been allocated within ten days (Sodemann, 2020), unfortunately, in other cases only a small fraction of moisture has been allocated after ten days. Nevertheless, we recognize that the tracking accuracy decreases with simulation time."

**Further comments**

- p. 2, l. 54:

"If the time step is chosen too large, real moisture transport may occur faster than the simulation grid and time step allow for (i.e., if the Courant number C =vΔt/Δx> 1). If the time step is taken too small, numerical diffusion will occur, meaning that moisture transport in the model will be faster than in the forcing data."

I am no expert in numerical modelling, but I believe that this phrasing is a bit misleading. As far as I know, numerical diffusion is primarily a consequence of the use of numerical schemes to represent partial differential equations as finite differences, which enables a numerical rather than analytical (and thus exact) solution. While particularly higher-order schemes provide very good approximations, even those are still associated with tiny errors, which, together with discretization errors and machine precision limitations, cause 'numerical diffusion'. I thus suggest to slightly rephrase the second quoted sentence to reflect that numerical diffusion becomes problematic for small timesteps (rather than implying the former is solely caused by the latter), because of course, if there are inaccuracies in the calculation of e.g. wind speeds due to the use of some advection scheme, this becomes exacerbated if these calculations are repeated more often (due to smaller timesteps).

Thank you for noting this. We simply removed the reference to numerical diffusion and now state (in lines 55-56) that "If the time step is taken too small, moisture transport in the model will be faster than in the forcing data."

- p. 4, l. 94:

"In general, atmospheric moisture tracking is achieved by [...] keeping track of how much of that moisture rains out **where** ."

To me, it reads better if the 'where' is moved to '... keeping track **where** and how much of that moisture rains out'.

Thank you; we agree and revised the text accordingly.

- p. 13, l. 411:

"Also here, the figures show that slower vertical mixing increases the area where rainfall depends on evaporation from the studied sources. However, with omega, the rainfall from the sources is more equally distributed within the footprints than without omega. In other words, with omega the footprints show a pattern that is less influenced by diffusion (Figs. 10, S44–S49)."

The first and second sentences make perfect sense to me, but not the third one. I suppose it is true that there are stronger contrasts in the top row (no omega) of e.g. Fig. 10 as compared to the bottom row (omega), or phrased differently, rainfall is more equally distributed with (than without) omega. But then, considering how the footprints with omega are more smooth, and diffusion (by definition) diminishes gradients, how are the omega-footprints less influenced by diffusion?

This is a good point. We agree that this was not a correct inference so we deleted the third sentence (which does not influence our Discussion and Conclusions).

- p. 14, l. 444:

"Also **,** the distances of moisture transport differed by several degrees for each of these models."

I believe a comma after 'Also' would help here.

We agree and added the comma.

- P. 15, l. 467:

[revised manuscript text omitted]

---

## Author Response (AR3)

Dear Louise Slater,

Thank you for your comments on our manuscript. We have incorporated all your suggestions as follows:

- Changed title to reflect the content, especially the sensitivity tests, better
- Added a paragraph to the discussion section to indicate that alternative models could be used and what would be necessary to properly do a model intercomparison.
- Changed the Github pages, note that that we changed the model name as well, thanks for these suggestions.

Kind regards,
Obbe Tuinenburg
Arie Staal

[revised manuscript text omitted]